# Kupffer cells control neonatal hepatic metabolism via Igf1 signaling

Nikola Makdissi[1,*], Daria J. Hirschmann[1,*], Aleksej Frolov[2,3,4], Inaam Sado[1], Bastian Bennühr[5], Fabian Nikolka[5], Jingyuan Cheng[6], Nelli Blank-Stein[1], Maria Francesca Viola[1], Mohamed Yaghmour[7], Philipp Arnold[8], Lorenzo Bonaguro[3,9], Matthias Becker[3,10], Christoph Thiele[7], Felix Meissner[6], Karsten Hiller[5], Marc D. Beyer[2,3,11] and Elvira Mass[1,‡]

## ABSTRACT

During perinatal development, liver metabolism is tightly regulated to ensure energy supply for the newborn. Before birth, glycogen is stored in hepatocytes and later metabolized to glucose, meeting neonatal energy demands. Shortly after birth, lipogenesis begins, driven by transcriptional activation of enzymes involved in fatty acid oxidation. These processes are thought to be largely regulated by systemic insulin and glucagon levels. However, the role of liver-derived local factors in neonatal hepatocyte metabolism remains unexplored. Kupffer cells (KCs), the liver's resident macrophages, colonize the fetal liver early in embryogenesis and support liver metabolism in adulthood. Yet whether KCs influence neonatal hepatocyte metabolism is unknown. Using conditional knockout mouse models targeting macrophages, we demonstrate that yolk sac-derived KCs play a crucial role in hepatocyte glycogen storage and function by regulating the tricarboxylic acid cycle, a role monocyte-derived KC-like cells cannot substitute. Newborn pups lacking yolk sac-derived KCs mobilize glycogen more rapidly, a process in part regulated by insulin-like growth factor 1 (Igf1) production. Our findings identify KCs as major source of Igf1, with local production essential for balanced hepatocyte metabolism at birth.

[1]Developmental Biology of the Immune System, Life & Medical Sciences (LIMES) Institute, University of Bonn, 53115 Bonn, Germany. [2]Immunogenomics & Neurodegeneration, Deutsches Zentrum für Neurodegenerative Erkrankungen (DZNE), 53127 Bonn, Germany. [3]Systems Medicine, Deutsches Zentrum für Neurodegenerative Erkrankungen (DZNE), 53127 Bonn, Germany. [4]Department of Microbiology and Immunology, The Peter Doherty Institute for Infection and Immunity, University of Melbourne, Melbourne, VIC 3000, Australia. [5]Department of Bioinformatics and Biochemistry, Braunschweig Integrated Centre of Systems Biology (BRICS), Technische Universität Braunschweig, 38106 Braunschweig, Germany. [6]Institute of Innate Immunity, Department of Systems Immunology and Proteomics, Medical Faculty, University of Bonn, 53127 Bonn, Germany. [7]Biochemistry & Cell Biology of Lipids, Life & Medical Sciences (LIMES) Institute, University of Bonn, 53115 Bonn, Germany. [8]Institute of Functional and Clinical Anatomy, Friedrich-Alexander-Universität Erlangen-Nürnberg (FAU), 91054 Erlangen, Germany. [9]Institute for Clinical Chemistry and Clinical Pharmacology, University Hospital Bonn, 53127 Bonn, Germany. [10]Modular High-Performance Computing and Artificial Intelligence, Deutsches Zentrum für Neurodegenerative Erkrankungen (DZNE), 53127 Bonn, Germany. [11]PRECISE Platform for Single Cell Genomics and Epigenomics, DZNE and University of Bonn and West German Genome Center, 53127 Bonn, Germany.

*These authors contributed equally to this work

‡Author for correspondence (elvira.mass@uni-bonn.de)

N.M., 0000-0002-9345-038X; D.J.H., 0009-0008-4538-4817; A.F., 0000-0002-4406-1206; M.F.V., 0000-0003-0660-9506; P.A., 0000-0003-3273-9865; L.B., 0000-0001-9675-7208; M.B., 0000-0002-7120-4508; C.T., 0000-0001-5161-2558; M.D.B., 0000-0001-9704-148X; E.M., 0000-0003-2318-2356

KEY WORDS: Liver development, Macrophage, Kupffer cell, Hepatocytes, Igf1

## INTRODUCTION

The liver is a highly versatile organ that shifts from being the main site of hematopoiesis during embryogenesis to one of the major metabolically active organs in the body after birth (Nakagaki et al., 2018). In neonates, the liver undergoes a metabolic adaptation to support postnatal energy demands (Gong et al., 2020; Li et al., 2023). Before birth, starting at embryonic day (E) 17, glycogen is stored in fetal hepatocytes (Tye and Burton, 1980). These stores are mobilized immediately after birth to supply glucose to maintain blood glucose levels during the initial hours prior to milk intake. Within just a few hours, this glycogenolysis phase shifts to active fatty acid oxidation (FAO) and engagement of the tricarboxylic acid (TCA) cycle, which, together with the supply of amino acids and essential co-factors, fuel gluconeogenesis to maintain glucose homeostasis as milk-derived dietary lipids become the primary energy source. Concurrently, neonatal hepatocytes shift away from glycolysis toward oxidative phosphorylation, maximizing ATP production to meet the high metabolic requirements of the newborn (Li et al., 2023). This metabolic switch ensures a steady glucose supply, which is essential for neonatal tissue homeostasis.

Neonatal liver metabolism is controlled by a combination of hormonal, genetic and environmental factors. The hormones insulin and glucagon play central roles in inducing glycogen breakdown and gluconeogenesis. After birth, glucagon levels rise sharply, stimulating glycogenolysis and gluconeogenesis, while insulin levels remain low until lactation begins. Lactation provides triglycerides (TGs), which are broken down into fatty acids and become the primary energy source for ATP production through FAO and the TCA cycle. Simultaneously, hepatocytes undergo maturation marked by the upregulation of key transcription factors and the expression of metabolically essential genes, which together support increased metabolic activity (Gong et al., 2020; Yang et al., 2023).

Before birth, the liver upregulates expression of the insulin receptor (Insr) (Gong et al., 2020) to respond to increasing insulin levels and the demand for FAO. In addition to insulin, insulin-like growth factor 1 (Igf1) is a natural ligand of the Insr. Igf1 is an important factor regulating growth and metabolism and is closely linked to insulin signaling pathways (Kineman et al., 2018, 2025). In the adult organism, hepatocytes are the primary source of circulating Igf1 (Kineman et al., 2018, 2025). In contrast, single-cell analyses of the fetal liver at E13.5 identified Kupffer cells (KCs), the resident hepatic macrophages, as a major local source of Igf1 (Tang et al., 2025). While mature hepatocytes do not express significant levels of the Igf1 receptor (Igf1r), its expression is detectable during the early stages of hepatocyte development (Waraky et al., 2016).

This suggests that during development KC-derived Igf1 may influence hepatocyte maturation or metabolic function through Insr/Igf1 signaling pathways. Recent studies further support this hypothesis by demonstrating that macrophage-derived Igf1 in the brain and gut plays a crucial role in tissue development (Yan et al., 2022; Rusin et al., 2024).

KCs originate from yolk sac-derived erythro-myeloid progenitors (EMPs) (Gomez Perdiguero et al., 2015). These EMPs differentiate into pre-macrophages (pMacs), which migrate to the liver and establish the early KC population as early as E10.25 (Mass et al., 2016), preceding the differentiation of hepatoblasts into hepatocytes. We have recently shown that KCs play a vital role in supporting fetal liver hematopoiesis, which peaks at E13.5-E14.5 (Kayvanjoo et al., 2024), underscoring the importance of macrophages in organ development and function (Mass et al., 2023). During adulthood, KCs are also known to regulate liver metabolism through insulin-like growth factor binding protein 7 (Igfbp7) (Morgantini et al., 2019). However, their role in neonatal hepatocyte maturation and metabolism remains unexplored.

In this study, we investigated the role of KCs and KC-derived factors in the development and function of neonatal hepatocytes. Using two KC depletion models and conditional Igf1 knockout mice, we found that KCs are important for proper metabolic function of neonatal hepatocytes. KC depletion accelerated hepatocyte maturation and disrupted metabolic regulation at the transcriptional level. This dysregulation was reflected functionally by reduced TCA cycle activity and diminished insulin responsiveness, resulting in reduced glycogen storage in hepatocytes. Notably, these functions were not restored by KC-like cells that repopulated the empty KC niche. Additionally, depletion of KC-derived Igf1 also led to a decreased glycogen content. Together, these findings reveal that yolk sac-derived KCs and their production of Igf1 are crucial for neonatal hepatocyte development and function.

## RESULTS
### Characterization of Kupffer cell depletion models
To address the role of KCs in neonatal hepatocyte metabolism, we employed two different mouse models that lead to the depletion of tissue-resident macrophages: *Tnfrsf11a*$^{Cre}$*; Spi1*$^{fl/fl}$ and *Tnfrsf11a*$^{Cre}$*; Csf1r*$^{fl/fl}$ (Fig. 1A, Fig. S1). Using the *Tnfrsf11a*$^{Cre}$, which targets pMacs and therefore all macrophages during development (Mass et al., 2016), we depleted either the transcription factor Spi1 or the surface receptor Csf1r, which are required for macrophage development and survival (Cox et al., 2021; Kayvanjoo et al., 2024; Jacome-Galarza et al., 2019). While we could confirm a complete depletion of F4/80$^{high}$CD11b$^{int}$ KCs at E14.5 in *Tnfrsf11a*$^{Cre}$*; Spi1*$^{fl/fl}$ livers (*KO*$^{Spi1}$) compared to littermates (*WT*$^{Spi1}$) (Kayvanjoo et al., 2024) (Fig. 1B,C; see Fig. S1B for gating strategy), the peak of depletion in *Tnfrsf11a*$^{Cre}$*; Csf1r*$^{fl/fl}$ embryos (*KO*$^{Csf1r}$) was detected at E12.5 (Fig. S1C,D). At birth [postnatal day (P) 0], the livers of *KO*$^{Spi1}$ and *KO*$^{Csf1r}$ animals were repopulated by CD11b$^{int}$F4/80$^{high}$ cells (Fig. 1D,E, Fig. S1E), with *KO*$^{Csf1r}$ pups even showing significantly increased macrophage numbers (Fig. S1E), a phenomenon that has been previously observed after the depletion of tissue-resident macrophages via blockade of Csf1r (Elmore et al., 2018).

To investigate whether the depleted KC niche would be repopulated by EMP-derived cells or by an alternative source, such as hematopoietic stem cell (HSC)-derived monocytes, we utilized a genetic labeling approach. In this context, it is important to note that, although KCs primarily arise from yolk sac-derived EMPs, HSC-derived monocytes can also serve as a source for macrophages under

certain conditions, including injury or inflammation (Mass et al., 2023; Schultze et al., 2019). To distinguish between HSC- and EMP-origin, we crossed the *KO*$^{Spi1}$ model to the *Ms4a3*$^{FlpO}$; *Rosa26*$^{FLF-tdTomato}$ model (Fig. 1F), which specifically labels cells originating from the definitive HSC wave, more specifically from granulocyte-monocyte progenitors (GMPs) that express *Ms4a3* (Liu et al., 2019; Huang et al., 2025). This allowed us to trace the contribution of GMP-derived monocytes in the repopulation process, which constituted ~80% of all Ly6C$^+$ monocytes in the fetal liver at P0 in both *WT*$^{Spi1}$ and *KO*$^{Spi1}$ mice (Fig. 1G). While the number of tdTomato (tdT)$^+$ (i.e. GMP-derived) CD11b$^{int}$F4/80$^{high}$ cells increased significantly from ~1% to 5-7%, a large fraction remained tdT$^-$ (Fig. 1G). This indicates that EMP-derived monocytes, EMP-derived pMacs residing in the fetal liver (Gomez Perdiguero et al., 2015; Stremmel et al., 2018; Mass et al., 2016), or other GMP-independent progenitors can readily differentiate into KC-like cells if the KC niche is empty. Of note, tdT$^+$ monocyte labeling was slightly reduced in *KO*$^{Spi1}$; *Ms4a3* livers (Fig. 1G), likely reflecting the recruitment of these cells to other tissues lacking tissue-resident macrophages.

To characterize phenotypic differences between KCs and KC-like cells, and to assess whether tdT$^+$ and tdT$^-$ populations share comparable phenotypic properties, we analyzed the *Tnfrsf11a*$^{Cre}$; *Spi1*$^{fl/fl}$ model crossed to *Ms4a3*$^{FlpO}$; *Rosa26*$^{FLF-tdTomato}$ (hereafter referred to as *WT*$^{Spi1}$; *Ms4a3* and *KO*$^{Spi1}$; *Ms4a3*) by flow cytometry. In *WT*$^{Spi1}$; *Ms4a3* livers, tdT$^+$ KCs expressed higher levels of CD45 (Ptprc), CD11b (Itgam) and F4/80 (Adgre1) than tdT$^-$ KCs, with CD11b and F4/80 showing a significant increase while expression of Tim4 (Timd4), Clec2 (Clec1b), Clec4f, Cx3cr1 and CD64 (Fcgr1) remained unchanged (Fig. 1H). In contrast, KC-like cells from *KO*$^{Spi1}$; *Ms4a3* livers displayed a distinct expression profile. Compared to *WT*$^{Spi1}$; *Ms4a3* controls, tdT$^-$ KC-like cells showed significantly elevated Tim4 and F4/80 expression, accompanied by reduced CD11b, Clec2 and Clec4f levels. A similar pattern was observed in tdT$^+$ KC-like cells from *KO*$^{Spi1}$; *Ms4a3* livers, particularly for Tim4, CD11b and Clec4f (Fig. 1H). *KO*$^{Spi1}$ mice analyzed without the fate-mapper background consistently showed reduced CD11b and increased Tim4 expression (Fig. S1F). Furthermore, in *WT*$^{Spi1}$; *Ms4a3* livers, KCs segregated into Vsig4$^+$ and Vsig4$^-$ populations irrespective of origin, whereas in the *KO*$^{Spi1}$; *Ms4a3* livers, the Vsig4$^+$ subset was almost completely absent (Fig. 1I). Together, these results indicate that ontogeny influences KC marker expression (CD11b, F4/80) in wild-type livers, whereas in *KO*$^{Spi1}$; *Ms4a3* animals the absence of yolk sac-derived KCs and their subsequent replacement by fetal liver progenitors introduces an additional layer shaping KC identity.

Building on the phenotypic differences observed between KCs and KC-like cells, we next examined whether these distinctions extended to the transcriptional level. Transcriptome analysis of sorted F4/80$^{high}$CD11b$^{int}$ KCs and KC-like cells from P0 *KO*$^{Spi1}$ and *WT*$^{Spi1}$ livers, respectively, showed that the freshly differentiated KC-like cells had an altered transcriptional profile (Fig. 2A), resulting in 1282 downregulated and 1463 upregulated differentially expressed genes (DEGs) (Fig. 2B, Table S1). Of note, the number of tdT$^+$ KCs in *WT*$^{Spi1}$; *Ms4a3* livers was very low, suggesting that the transcriptomic signature of bona fide yolk sac-derived KCs dominates in these samples. Therefore, we did not distinguish between tdT$^+$ and tdT$^-$ subsets for subsequent analyses.

Gene set enrichment analysis (GSEA) indicated that genes falling into terms such as 'protein secretion', 'angiogenesis' and 'oxidative phosphorylation' were enriched in *KO*$^{Spi1}$ KC-like cells, while genes belonging to 'inflammatory response', 'heme metabolism', 'interferon alpha response' and 'interferon gamma response' were

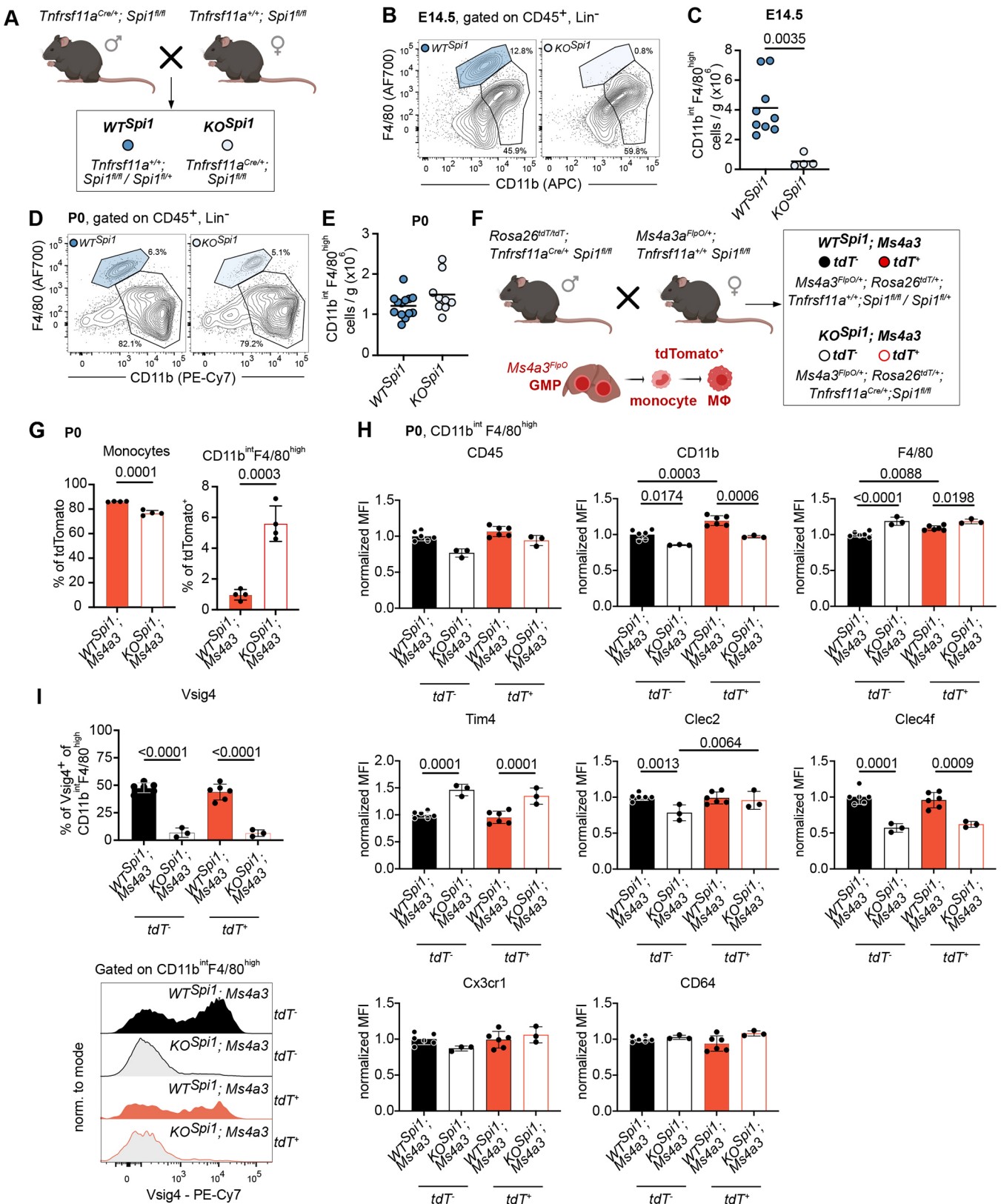

**Fig. 1.** See next page for legend.

downregulated (Fig. 2C, Table S2). Upregulated DEGs falling into the term 'oxidative phosphorylation' included *Slc25a3*, *Ndufs2* and *Idh1*, whereas *KO^{Spi1}* KC-like cells downregulated genes such as *Tnf*, *Il6*, *Il1a*, *Il1b* and *Cxcl10* falling into the term 'inflammatory response' (Fig. 2D). These findings, along with the altered surface expression of macrophage markers, demonstrate that the newly formed KC-like cells possess a distinct molecular composition compared to bona fide yolk sac-derived KCs. In summary, the

**Fig. 1. Characterization of the $KO^{Spi1}$ mouse model.** (A) Breeding scheme to produce $KO^{Spi1}$ and littermate controls ($WT^{Spi1}$). Created in BioRender by Mass, E., 2025. https://BioRender.com/jvsfc8p. This figure was sublicensed under CC-BY 4.0 terms. (B) Representative flow cytometry plots of $WT^{Spi1}$ and $KO^{Spi1}$ fetal livers at E14.5 showing efficient depletion of KCs. (C) Quantification of total $WT^{Spi1}$ and $KO^{Spi1}$ KC numbers at E14.5. Circles represent individual mice. $n$=4-9 per genotype from 3 independent litters. Unpaired Student's $t$-test. (D) Representative flow cytometry plots of $WT^{Spi1}$ and $KO^{Spi1}$ livers at P0 showing repopulation of the empty KC niche. (E) Quantification of total $WT^{Spi1}$ and $KO^{Spi1}$ KC numbers at P0. Circles represent individual mice. $n$=9-13 per genotype from 5 independent litters. (F) Breeding scheme to produce $KO^{Spi1}$ mice and littermate controls in combination with the granulocyte-monocyte progenitor (GMP) fate-mapping model $Ms4a3^{FlpO}$. Labeled GMPs differentiate into monocytes and subsequently into macrophages (MΦ). (G) Quantification of fate-mapped Ly6C$^+$ monocytes, KCs and KC-like cells in $WT^{Spi1}$ and $KO^{Spi1}$ mice at P0. Circles represent individual mice. $n$=4 per genotype from 2 independent litters. Unpaired Student's $t$-test. (H) Normalized expression of surface receptor markers on fate-mapped KCs and KC-like cells from $WT^{Spi1}$; $Ms4a3$ and $KO^{Spi1}$; $Ms4a3$ mice at P0. $n$=3-6 per genotype from 2 independent litters. Mixed-effects model (REML) test with Šidák's correction for multiple comparison test. (I) Quantification of Vsig4$^+$ macrophages in $WT^{Spi1}$; $Ms4a3$ and $KO^{Spi1}$; $Ms4a3$ mouse livers at P0. $n$=3-6 per genotype from 2 independent litters. Mixed-effects model (REML) test with Šidák's correction for multiple comparison test. Data are shown as mean±s.d. MFI, median fluorescence intensity.

$KO^{Spi1}$ and $KO^{Csf1r}$ models represent two complementary KC-depletion models that allow the role of macrophage ontogeny in tissue development and homeostasis to be studied.

## KC depletion leads to decreased glycogen storage at birth

After establishing the KC-depletion models, we investigated whether the replacement of yolk sac-derived, long-lived KCs by KC-like cells could affect neonatal hepatic metabolism. The liver plays an essential role in energy conversion, maintaining blood glucose homeostasis in neonates by storing glycogen during embryogenesis, which is subsequently metabolized through extensive glycogenolysis after birth (Li et al., 2023). Therefore, we assessed glycogen levels in P0 $WT^{Spi1}$ and $KO^{Spi1}$ livers using Periodic acid-Schiff (PAS) staining (Fig. 3A). Quantification of the staining indicated that $KO^{Spi1}$ livers stored less glycogen than their littermate controls (Fig. 3B,C). For a more quantitative analysis, we used a colorimetric assay, which corroborated our histological results, showing that $KO^{Spi1}$ livers had approximately half the glycogen concentration of $WT^{Spi1}$ livers (Fig. 3D). Moreover, transmission electron microscopy images clearly demonstrated a reduction in glycogen content in $KO^{Spi1}$ compared to $WT^{Spi1}$, where the glycogen appears more abundant and densely packed (Fig. 3E). We further confirmed a significantly reduced glycogen content in the $KO^{Csf1r}$ livers compared to $WT^{Csf1r}$ using a colorimetric assay (Fig. S1G). Collectively, our findings suggest that the replacement of depleted KCs by KC-like cells leads to altered hepatocyte metabolism at birth, characterized by significant changes in glycogen storage.

## Hepatocytes undergo a metabolic shift after Kupffer cell depletion

To understand whether hepatocytes are altered functionally due to the lack of KCs, we first performed single-cell RNA-sequencing (scRNA-seq) analysis of livers from P0 $WT^{Spi1}$ and $KO^{Spi1}$ (Fig. S2A,B). We identified hepatocytes based on lineage markers (Fig. S2B). After identifying all DEGs between $WT^{Spi1}$ and $KO^{Spi1}$, we ranked these genes based on average $\log_2$ fold-change and used the top 100 up- or downregulated genes for gene ontology (GO) enrichment analysis (Table S3). The results of this analysis showed a significant upregulation of 'lipid transport', 'ATP metabolic process', 'pyruvate metabolic process' and 'response to insulin', and downregulation of 'fatty acid metabolic process' and 'gluconeogenesis' terms in $KO^{Spi1}$ hepatocytes (Fig. 4A).

Given the dysregulation of metabolic pathways at the transcriptional level and the accelerated glycogen degradation, we aimed to determine whether the absence of KCs affects hepatocyte maturation in $KO^{Spi1}$ animals. To this end, we subclustered all hepatocytes, which resulted in three distinct clusters (Fig. 4B). We examined the developmental trajectory of hepatocytes using trajectory inference with Monocle3. To rank cells on a pseudotemporal scale, we performed pseudotime analysis with a cell from cluster 0 as root node, because cells in this cluster expressed the lowest levels of hepatocyte maturation markers, such as $Alb$, $Aldob$, $Ahsg$ and $Ttr$ (Gong et al., 2020). Thus, in this analysis, cluster 0 represents the immature state, and progresses to cluster 2, which represents the most mature state (Fig. 4C). As the distribution of cells among cluster 2 was altered in $KO^{Spi1}$ hepatocytes (Fig. 4B), we ranked genes expressed by cluster 2 from $WT^{Spi1}$ and $KO^{Spi1}$ hepatocytes based on average $\log_2$ fold-change and used the top 100 up- or downregulated genes for GO enrichment analysis. This analysis revealed that distinct genes falling into the same terms were either upregulated in $WT^{Spi1}$ or in $KO^{Spi1}$ hepatocytes (Fig. S2C). Common terms included 'fatty acid metabolic process' and 'gluconeogenesis' with genes such as $Lpin1$ and $Lpin2$, which are important for TG biosynthesis, showing a reduced expression in $KO^{Spi1}$ hepatocytes, while other genes related to fatty acid metabolic processes, such as $Apoa1$ and $Lpl$, were expressed at higher levels in $KO^{Spi1}$ hepatocytes (Fig. S2C). These data suggest that $KO^{Spi1}$ hepatocytes, despite belonging to the most mature cluster, shifted their metabolic functionality compared to $WT^{Spi1}$ hepatocytes. To further address this transcriptional shift, we performed a density distribution (Fig. 4D) and an empirical cumulative distribution analysis (Fig. 4E), focusing on the pseudotime interval 12-15. These analyses revealed that $KO^{Spi1}$ hepatocytes progressed more rapidly toward the mature state. We corroborated this by performing a pseudo-bulk analysis of DEGs in cluster 2: the upregulation of hepatocyte maturation markers, such as $Alb$, $Aldob$, $Ahsg$ and $Ttr$ (Gong et al., 2020), in $KO^{Spi1}$ hepatocytes compared to $WT^{Spi1}$ (Fig. 4F) further supported the hypothesis that KC deficiency during embryogenesis promotes hepatocyte maturation on the transcriptional level.

To determine whether $KO^{Spi1}$ hepatocytes exhibit increased metabolic activity due to their accelerated transcriptional maturation, we performed 5-h [U-$^{13}C_6$]-glucose tracing experiments on freshly isolated livers from $WT^{Spi1}$ and $KO^{Spi1}$ mice. Normalized metabolite abundance showed significantly higher levels of pyruvate and lactate in $KO^{Spi1}$ livers, while alanine, aspartate, citrate and malate remained comparable between conditions (Fig. 4G). [U-$^{13}C_6$]-labeled metabolites also showed increased isotope incorporation into pyruvate and lactate (Fig. 4H,I); however, glycolytic acetyl-CoA incorporation into citrate (M2) was significantly lower in $KO^{Spi1}$ livers, while contribution to malate was similar across genotypes (Fig. 4H,I). Collectively, these findings suggest increased glycolytic flux, reduced pyruvate dehydrogenase (PDH) activity, and altered carbon flow into the TCA cycle in $KO^{Spi1}$ livers. Moreover, hepatocytes showed decreased reactive oxygen species production, as measured by MitoSOX staining (Fig. 4J). The MitoSOX signal serves as a proxy for mitochondrial respiration measurement and, therefore, indicates reduced TCA cycle activity, possibly due to the decreased PDH flux. Of note, measurement of MitoTracker green fluorescence (Fig. 4K), indicative of mitochondrial quantity, and quantification of the cristae area coverage in mitochondria using transmission electron microscopy (Fig. 4L) did not reveal any changes between $WT^{Spi1}$ and $KO^{Spi1}$ livers. Additionally, high-dimensional

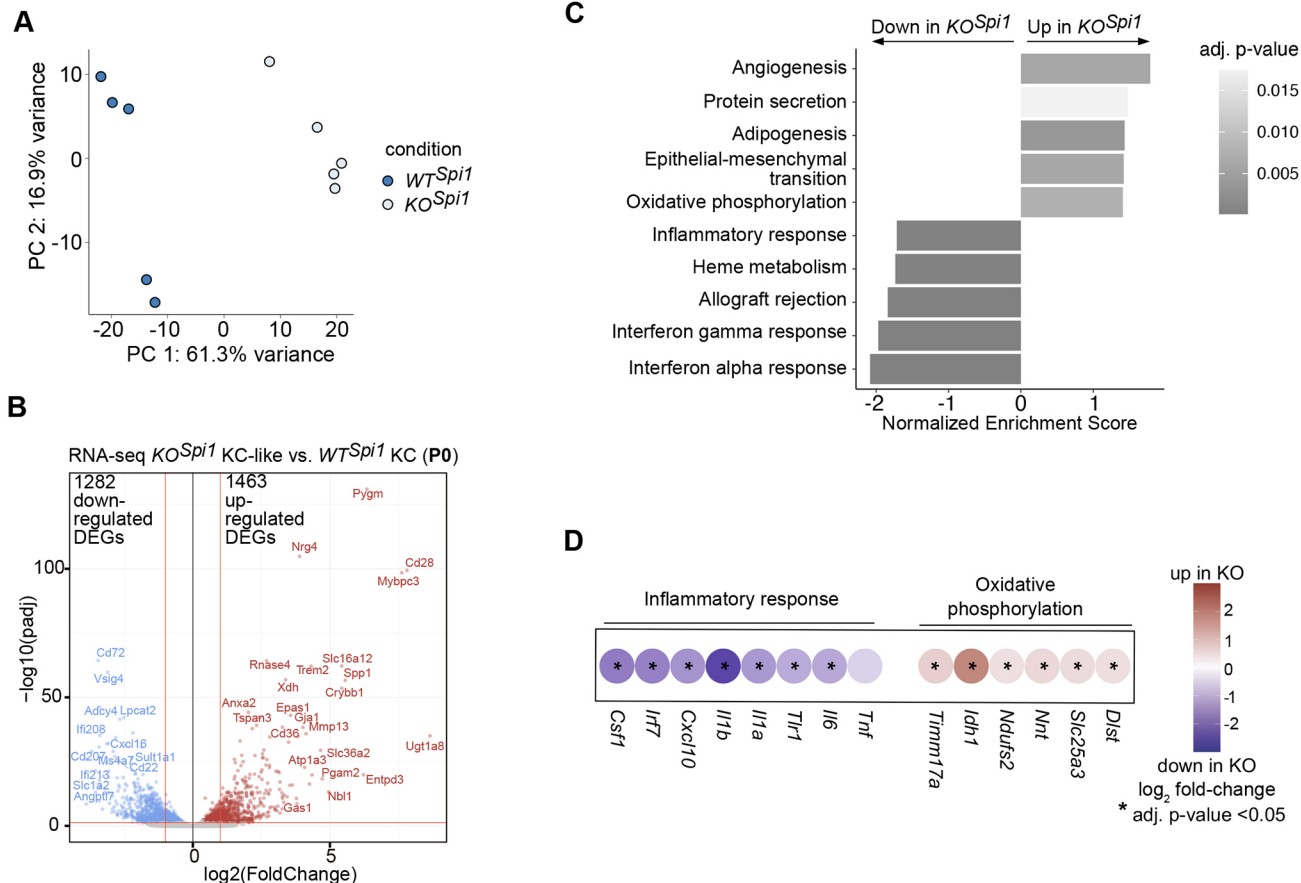

**Fig. 2. Transcriptional alterations in KC-like cells from $KO^{Spi1}$ neonates.** (A) Principal component analysis of the $WT^{Spi1}$ and $KO^{Spi1}$ KC and KC-like macrophages, respectively. $n$=5 per genotype from 3 independent litters. (B) Volcano plot showing DEGs comparing KCs from $WT^{Spi1}$ and KC-like cells from $KO^{Spi1}$ mice at P0. Differential expression was tested using DESeq2 on raw counts, log2FoldChange=1.3, $P$-adjust=0.05. (C) GO enrichment analysis of differentially up- and downregulated genes comparing KCs from $WT^{Spi1}$ and $KO^{Spi1}$ mice at P0. (D) Selected genes from the terms 'oxidative phosphorylation' and 'inflammatory response' from C. Dot plot indicates upregulated (red) and downregulated (purple) genes when comparing $KO^{Spi1}$ to $WT^{Spi1}$. Differential expression was tested using DESeq2 on raw counts. Asterisks indicate adjusted $P$-value<0.05.

flow cytometry of hepatocytes did not detect significant changes in the abundance of various nutrient transporters [GLUT1 (SLC2A1), CD36, CD98 (SLC3A2)] and core metabolic enzymes [PKM, G6PD, SDHA, ATP5a, CPT1a, ACC1 (ACACA)] (Fig. S2D-F). These data confirm that the altered metabolic activity arises from changes in the metabolic response of hepatocytes rather than from impairments in mitochondrial health. In summary, our findings from metabolic tracing experiments indicate that $KO^{Spi1}$ hepatocytes have enhanced glycolytic activity and increased production of pyruvate and lactate, yet exhibit reduced TCA cycle engagement as evidenced by lower citrate incorporation.

To further explore whether the changes in PDH flux and TCA cycle activity impact lipid metabolism, we performed a comprehensive lipid analysis using mass spectrometry. Overall, no significant changes were found in lipid classes across both KO models (Fig. S3A,B). However, we observed a reduction in the most abundant TG species, including TG(50:2), TG(52:2) and TG(52:3), in both $KO^{Spi1}$ and $KO^{Csf1r}$ livers (Fig. 4M, Fig. S3C), indicating reduced synthesis of these specific lipid types. Collectively, our data demonstrate the requirement of KCs for proper neonatal hepatocyte function, as the presence of KC-like macrophages leads to a metabolic dysregulation, with a preference for lactate production over TCA cycle and lipid synthesis. This shift may also explain the need for increased consumption of glycogen to maintain ATP levels, leading to faster glycogen depletion (see Fig. 3).

## KC-derived Igf1 controls glycogen homeostasis in neonatal hepatocytes

To identify KC-dependent pathways potentially driving the metabolic shift and increased glycogen demand in hepatocytes, we used the scRNA-seq data from $WT^{Spi1}$ and $KO^{Spi1}$ livers and performed CellChat analysis, focusing on the interaction between KCs and hepatocytes (Fig. 5A). The analysis revealed several downregulated pathways in KC-like macrophages from $KO^{Spi1}$ animals, particularly those involved in regulating cellular metabolism, including the visfatin, IGF and SEMA6 pathways (Kang et al., 2018; Nakanishi et al., 2024; Revollo et al., 2007; Kineman et al., 2018, 2025; Fellinger et al., 2023). Notably, visfatin (also known as the extracellular form of Nampt) has been shown to regulate insulin secretion (Revollo et al., 2007). To investigate whether KC-like macrophages from $KO^{Spi1}$ livers interact uniformly with all hepatocyte populations or if specific hepatocyte clusters are more affected, we analyzed ligand–receptor pairs to explore these dynamics (Fig. 5B). Our analysis revealed that Nampt-Insr interactions had the highest probability of interaction between $WT^{Spi1}$ KCs and all three hepatocyte clusters, but were markedly reduced or absent in $KO^{Spi1}$ livers (Fig. 5B). Igf1-Igf1r signaling was highest between $WT^{Spi1}$ KCs and hepatocyte cluster 2 (Hepa2), but showed reduced communication probability between KCs and Hepa2 in $KO^{Spi1}$ livers. Similar behavior was observed for the ligand–receptor pair Sema6a-Plxna2 (Fig. 5B). Although not listed

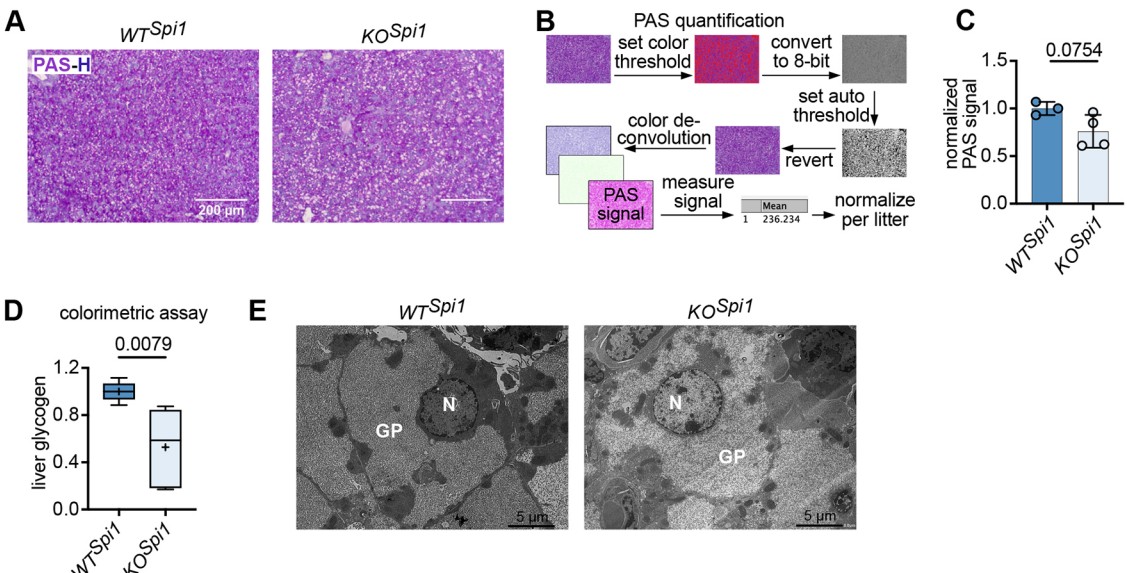

**Fig. 3. $KO^{Spi1}$ hepatocytes contain less glycogen storage at birth.** (A) Representative PAS (purple)-Hematoxylin (H, blue) staining of $WT^{Spi1}$ and $KO^{Spi1}$ livers at P0. Scale bars: 200 µm. (B) Scheme indicating how PAS staining intensity was quantified using Fiji. (C) Normalized PAS signal intensity of $WT^{Spi1}$ and $KO^{Spi1}$ livers at P0. Circles represent individual mice. $n$=3-4 per genotype from 2 independent litters. Paired Student's $t$-test. Data are shown as mean±s.d. (D) Glycogen levels measured on whole liver lysates of $WT^{Spi1}$ and $KO^{Spi1}$ P0 livers. $n$=5 per genotype from 3 independent litters. Values were normalized per litter. The whiskers represent the 5-95% percentile, the box extends from the 25th to 75th percentiles and the horizontal line represents the median. Cross indicates the mean. Mann–Whitney test. (E) Representative transmission electron micrograph from $WT^{Spi1}$ and $KO^{Spi1}$ livers at P0. $n$=3 per genotype from 2 independent litters. GP, glycogen particle; N, nucleus. Scale bars: 5 µm.

among the main signaling pathways altered between $WT^{Spi1}$ and $KO^{Spi1}$ animals (Fig. 5A), we predicted a potential interaction of Tnf and Tnfrsf1a in $WT^{Spi1}$ livers based on *Tnf* gene expression in KCs and *Tnfrsf1a* gene expression in all hepatocyte clusters (Fig. 5B). This interaction was absent in $KO^{Spi1}$ livers (Fig. 5B). This further supports the hypothesis that neonatal yolk sac-derived KCs play a role in regulating hepatocyte metabolism, as KC-derived Tnf is known to control lipid uptake in hepatocytes (Diehl et al., 2020).

Next, we analyzed the pathways directly associated with Insr and Igf1r in more detail. Network centrality analysis in $WT^{Spi1}$ KCs and $KO^{Spi1}$ KC-like cells and hepatocyte clusters suggested that $WT^{Spi1}$ KCs were the primary senders, mediators and influencers of the IGF signaling pathway. In contrast, $KO^{Spi1}$ KC-like cells exhibited reduced sender activity, with the least mature hepatocyte cluster 0 (Hepa0) emerging as a prominent sender (Fig. 5C). Further analysis of the visfatin (Nampt) signaling pathway revealed that the most mature cluster Hepa2 in $KO^{Spi1}$ livers showed a diminished role in receiving signals via Insr. Additionally, Hepa2 in $KO^{Spi1}$ livers showed diminished activity as mediator and influencer to its counterpart in $WT^{Spi1}$ animals (Fig. 5C). Although Nampt does not directly bind Insr, Igf1 is known to interact with Insr homodimers and Insr/Igf1r heterodimers. Both ligands activate downstream Insr signaling pathways, thereby regulating cellular metabolism (Saddi-Rosa et al., 2010; Kineman et al., 2018). To further investigate differential regulation of this pathway, we analyzed bulk RNA-seq data from P0 $WT^{Spi1}$ KCs and $KO^{Spi1}$ KC-like cells, as well as sorted hepatocytes from the same animals. This analysis revealed that *Igf1* gene expression was lower in $KO^{Spi1}$ KC-like cells compared to $WT^{Spi1}$ KCs, although this difference did not reach statistical significance, and was present at relatively low levels in hepatocytes (Fig. 5D). These findings support the notion that Igf1-dependent signaling is altered in $KO^{Spi1}$ animals, likely reflecting paracrine regulatory mechanisms.

Although these transcriptional data suggest altered expression of Igf1 and Insr pathway components, they do not fully reflect the dynamic regulation of signaling activity, which depends not only on gene expression but also on ligand availability and receptor phosphorylation status. In the neonatal liver, hepatocytes sense insulin via Insr to promote glucose uptake, whereas glucagon stimulates glycogen breakdown and glucose release (Li et al., 2023; Nevado et al., 2006). To determine whether altered systemic hormone levels contribute to the metabolic shift observed in $KO^{Spi1}$ hepatocytes, we measured plasma insulin and glucagon concentrations at P0. However, neither insulin nor glucagon levels differed between $KO^{Spi1}$ and $WT^{Spi1}$ animals (Fig. 5E,F), indicating that systemic hormonal changes are unlikely to account for the observed metabolic phenotype.

Next, we employed a quantitative phospho-proteomics approach and compared whole livers of $WT^{Spi1}$ and $KO^{Spi1}$ littermates at P0. To complement our findings from the scRNA-seq data, we specifically focused on glucose- and insulin-related signaling pathways. Enrichment analysis of downregulated phosphorylation sites comparing $KO^{Spi1}$ and $WT^{Spi1}$ livers (Table S4) showed that sites involved in 'glucose metabolic process', 'Igf1 binding', 'Igf1 signaling pathway' and 'insulin binding' were less phosphorylated. Conversely, sites involved in 'cellular response to glucose stimulus' showed an increased phosphorylation in $KO^{Spi1}$ livers compared to $WT^{Spi1}$ (Fig. 5G). Notably, downregulated phosphorylation sites in $KO^{Spi1}$ livers included key sites on insulin receptor substrate 1 (Irs1) – S307, S318 and S526 (Table S4) – defined as direct downstream activation sites of the Insr (Humphrey et al., 2013; Müssig et al., 2005; Rui et al., 2001; Parker et al., 2015). Additionally, the downregulated GO term 'insulin receptor complex' included the site S1340 on Insr, which is phosphorylated upon insulin stimulation (Parker et al., 2015). In summary, our findings suggest that KC-like macrophages in $KO^{Spi1}$ livers are unable to activate the signaling pathway downstream of Igf1 and Insr in neonatal hepatocytes as efficiently as bona fide KCs, potentially contributing to the observed depletion of hepatic glycogen stores.

Previous studies have shown that macrophages, especially during the early postnatal period, are major producers of Igf1

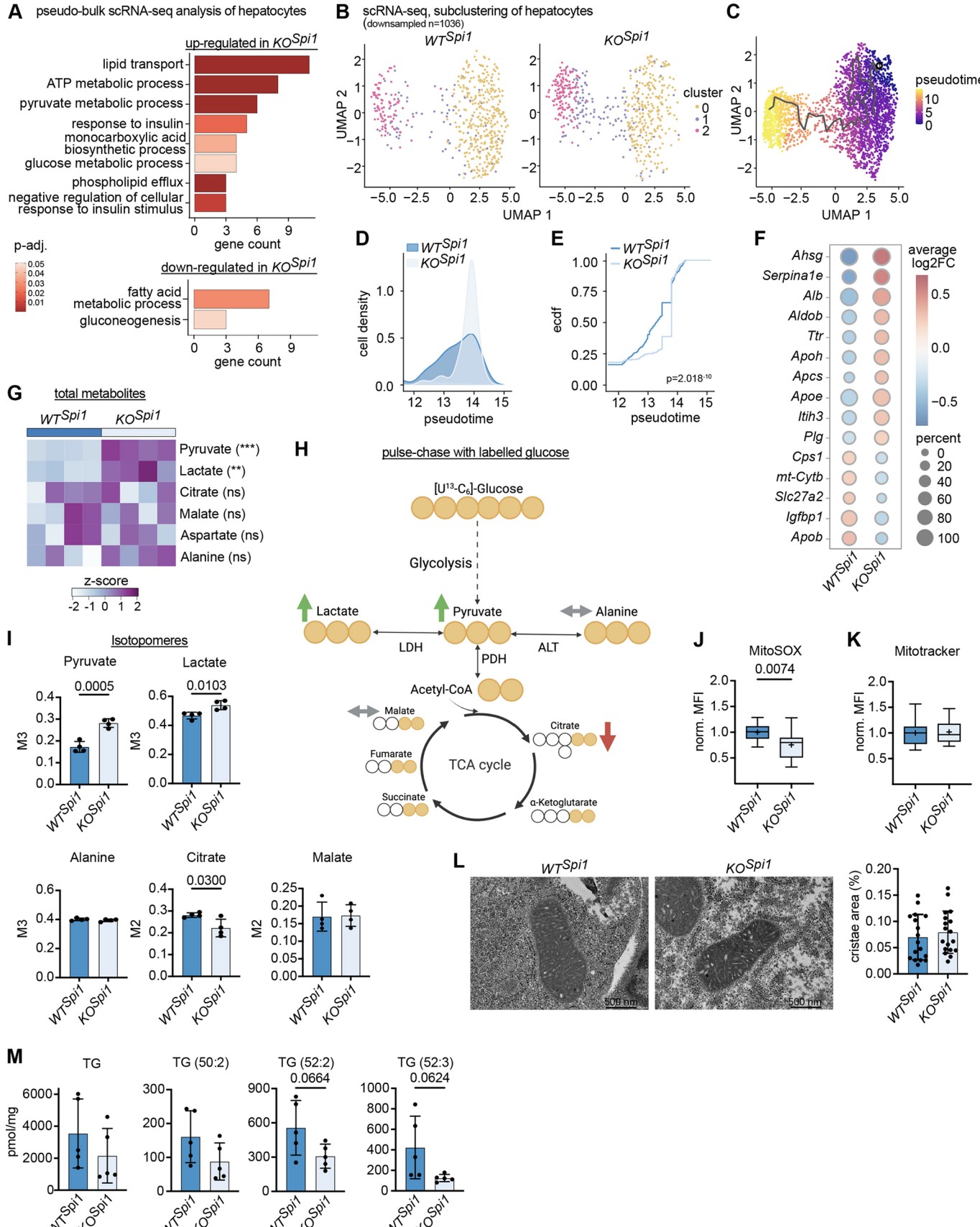

Fig. 4. See next page for legend.

**Fig. 4. $KO^{Spi1}$ hepatocytes exhibit an altered metabolism at birth following Kupffer cell depletion.** (A) Pseudo-bulk RNA-seq analysis of hepatocytes showing GO enrichment for up- and downregulated pathways in $KO^{Spi1}$ at P0. (B) UMAP analysis of P0 $WT^{Spi1}$ and $KO^{Spi1}$ hepatocytes from the scRNA-seq dataset. (C) Pseudotime trajectory analysis of the hepatocytes cluster using Monocle3. (D) Density distribution analysis of hepatocytes within the pseudotime interval 12-15. (E) Empirical cumulative distribution function (ecdf) analysis of hepatocytes within the pseudotime interval 12-15. (F) Pseudo-bulk analysis of DEGs in hepatocyte cluster 2. Dot plot shows $\log_2$ fold-change (color) and percentage of cells expressing each gene (dot size). (G) Normalized total metabolite abundance in of $WT^{Spi1}$ and $KO^{Spi1}$ livers following [U-$^{13}$C$_6$]-glucose tracing at P0. $n=4$ per genotype from 2 independent litters. Unpaired Student's $t$-test. **$P<0.01$, ***$P<0.001$. ns, not significant ($P>0.05$). (H) Schematic of the possible metabolite labeling patterns due to the incorporation of the [U-$^{13}$C$_6$]-glucose tracer. Created in BioRender by Hiller, K., 2025. https://BioRender.com/qo2etjn. This figure was sublicensed under CC-BY 4.0 terms. (I) Fractional enrichment of labeled metabolites following [U-$^{13}$C$_6$]-glucose tracing at P0. $n=4$ per genotype from 2 independent litters. Unpaired Student's $t$-test. (J,K) Normalized median fluorescence intensity (MFI) of MitoSOX Red (J) and MitoTracker Green (K) in hepatocytes at P0. $n=13$-14 per genotype from 5 independent litters. The whiskers represent the 5-95% percentile, the box extends from the 25th to 75th percentiles and the horizontal line represents the median. Cross indicates the mean. Unpaired Student's $t$-test. (L) Representative transmission electron micrographs of mitochondria of $WT^{Spi1}$ and $KO^{Spi1}$ livers (left), along with the quantified cristae area coverage (right). $n=17$-18 mitochondria per genotype from 2 independent litters. Scale bars: 500 nm. (M) The abundance of total triacylglycerol (TG) and its subspecies in $WT^{Spi1}$ and $KO^{Spi1}$ livers. $n=5$-6 per genotype from 3 independent litters. Unpaired Student's $t$-test. All bar plots are presented as mean±s.d.

(Pridans et al., 2018; Yan et al., 2022). Furthermore, analysis of the developmental atlas dataset from Qiu et al. (2024) revealed that, among liver-resident cells, KCs exhibit the highest *Igf1* expression, identifying them as the predominant producers during development (Fig. 6A). To address whether KC-derived Igf1 directly controls glycogenolysis, we generated the macrophage-specific Igf1 knockout mouse model *Tnfrsf11a$^{Cre}$; Igf1$^{fl/fl}$* (Fig. 6B). Conditional knockout mice ($KO^{Igf1}$) exhibited significantly reduced Igf1 levels, with liver Igf1 levels averaging around two-thirds and serum Igf1 levels around half of those observed in littermate controls ($WT^{Igf1}$; Fig. 6C,D). This reduction was accompanied by lower glycogen levels in the liver at P0, demonstrated by quantitative colorimetric assay (Fig. 6E) and further confirmed by transmission electron microscopy (Fig. 6F-H).

To further assess how KC-derived Igf1 influences hepatic glucose utilization, we performed stable-isotope tracing in neonatal livers from $WT^{Igf1}$ and $KO^{Igf1}$ mice incubated *ex vivo* with [U-$^{13}$C$_6$]-glucose for 5 h. In contrast to the increased glycolytic flux and reduced PDH activity observed in $KO^{Spi1}$ livers, loss of macrophage-derived Igf1 alone did not perturb central carbon metabolism. Isotopomer distribution and total metabolite abundances remained comparable between $WT^{Igf1}$ and $KO^{Igf1}$ livers, indicating that glycolytic and PDH fluxes were largely preserved (Fig. 6I,J).

We next tested whether exogenous Igf1 could modulate these pathways. Supplementation of recombinant Igf1 had no effect on $WT^{Igf1}$ livers but decreased glycolytic flux and enhanced PDH activity in $KO^{Igf1}$ livers, reflected by reduced M3-pyruvate and increased M2-malate with similar trends in M2-citrate labeling (Fig. 6J). Together, these data indicate that Igf1 signaling in wild-type neonatal livers operates near maximal capacity, rendering further stimulation ineffective, whereas livers lacking KC-derived Igf1 remain responsive to Igf1. Thus, acute restoration of Igf1 signaling in this context reorients hepatocyte metabolism toward oxidative pathways, the inverse of the glycolytic shift induced by KC depletion, demonstrating that loss and gain of Igf1 signaling exert opposing effects on neonatal hepatic metabolism.

In summary, these findings demonstrate that KC-derived Igf1 is an important regulator of neonatal hepatic glycogen and glucose metabolism. Its developmental loss disrupts glycogen storage and shifts hepatocytes toward glycolysis, whereas its acute re-introduction promotes oxidative metabolism through enhanced PDH activity, underscoring the central role of KC–hepatocyte cross-talk in establishing metabolic homeostasis after birth.

## DISCUSSION

Here, we show that metabolic function of neonatal hepatocytes depends on the early presence and co-development of yolk sac-derived KCs. Depletion of KCs during embryogenesis using two different models ($KO^{Spi1}$ and $KO^{Csf1r}$) led to a replenishment of the empty niche by KC-like macrophages. However, this replacement was not sufficient to support normal hepatocyte function, as evidenced by reduced hepatic glycogen storage in both models. These findings indicate that the timely co-development of EMP/pMac-derived KCs from the onset of liver organogenesis is essential and cannot be functionally compensated for by perinatally recruited KC-like cells.

This dependency reflects the unique characteristics of the neonatal liver environment. At birth, neither hepatocyte zonation nor the mature KC niche, characterized by extensive interaction with liver sinusoidal endothelial cells and hepatic stellate cells, is yet established (Bonnardel et al., 2019; Araujo David et al., 2024). Consequently, the macrophage–hepatocyte crosstalk during this developmental window differs fundamentally from the interactions observed in adult tissues. Early KC-derived signals appear to be indispensable to guide hepatocyte maturation before the tissue architecture and growth factor production by other liver-resident cells are fully established.

Using *Ms4a3* promoter-based fate-mapping in combination with the $KO^{Spi1}$ model, we confirmed an increased influx of GMP-derived monocytes into the KC niche. However, most KC-like macrophages did not originate from definitive HSCs. Recent studies suggest long-term (LT)-HSC-independent hematopoiesis in the fetal liver (Kobayashi et al., 2023; Yokomizo et al., 2022), which may partially account for the unlabeled cells in our model, as Ms4a3 only labels GMPs stemming from definitive LT-HSCs (Liu et al., 2019). Alternatively, EMP-derived monocytes, which are readily found in the fetal liver (Gomez Perdiguero et al., 2015), and which remain Ms4a3 negative, may be the source of new macrophages in the liver. Thus, our combined KO/fate-mapping approach highlights the so-far elusive *in vivo* potential of HSC-independent progenitors generating macrophages and the need to develop novel mouse models that can discriminate EMP- from HSC-derived monocytes.

KCs are among the first tissue-specific macrophages that can be detected in the developing embryo, as early as E10.25 (Mass et al., 2016). Thus, hepatocytes, which begin differentiating at E13.5 (Yang et al., 2017) and start glycogen storage at E17 (Tye and Burton, 1980), are continuously interacting with KCs, which only begin to migrate to the sinusoids at 1 week of age (Araujo David et al., 2024). This direct cell communication positions KCs as important influencers of hepatocyte maturation and function. Using a combination of scRNA-seq, phospho-proteomics and a conditional knockout of Igf1 in macrophages ($KO^{Igf1}$), we identify KCs as active regulators of hepatocyte function. Neonatal KCs produce Igf1, which supports TCA activity and glucose metabolism in hepatocytes before hepatocytes themselves begin producing Igf1.

Developmental reference datasets provide independent support for this conclusion. Analyses of single-cell atlases of the fetal and neonatal

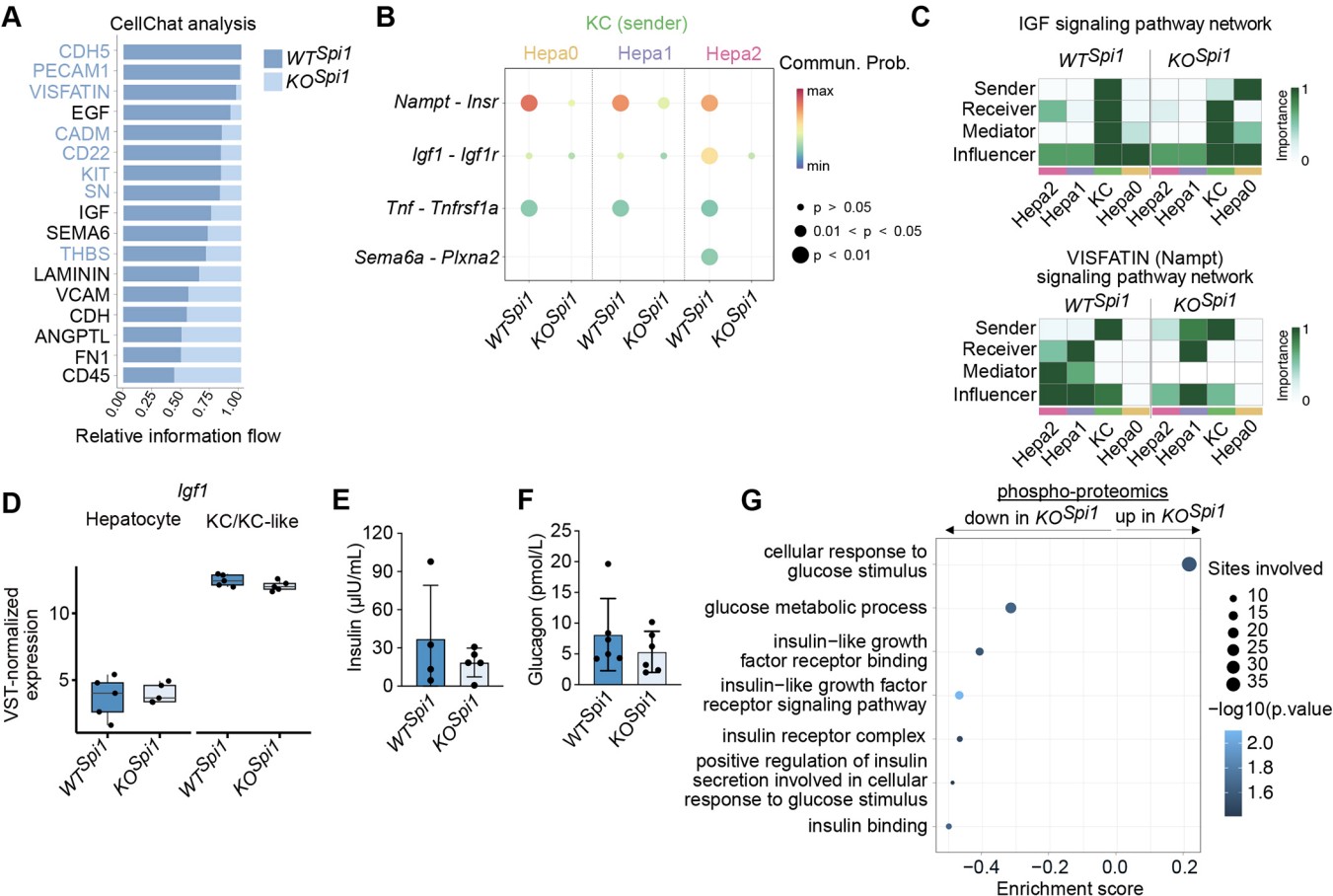

**Fig. 5. KC-hepatocyte crosstalk is altered in P0 *KO^Spi1* livers.** (A) Bar plot showing the relative information flow of between *WT^Spi1* and *KO^Spi1* of inferred cell–cell communication using CellChat. (B) Comparison of the significant ligand–receptor pairs between *WT^Spi1* and *KO^Spi1*, which contribute to the signaling from KCs to the hepatocyte clusters. (C) Heatmap showing the relative importance of KC and hepatocyte clusters as sender, receiver, mediator and influencer, based on the computed four network centrality measures of IGF (top) and visfatin (bottom) signaling. (D) Box plot of variance stabilizing transformation-normalized *Igf1* expression in hepatocytes and macrophages in *WT^Spi1* and *KO^Spi1* mice at P0. *n*=5 per genotype from 3 independent litters. Differential expression was tested using DESeq2 on raw counts. The whiskers represent the 5-95% percentile, the box extends from the 25th to 75th percentiles and the horizontal line represents the median. (E) Serum insulin levels measured by ELISA on *WT^Spi1* and *KO^Spi1* at P0. *n*=4-5 per genotype from 4 independent litters. Bar plot presented as mean±s.d. Unpaired Student's *t*-test. (F) Serum glucagon levels measured by ELISA on *WT^Spi1* and *KO^Spi1* at P0. *n*=6 per genotype from 3 independent litters. Bar plot presented as mean±s.d. Unpaired Student's *t*-test. (G) Enrichment analysis of downregulated phosphorylation sites showing the decreased and increased phosphorylation in *KO^Spi1* liver compared to *WT^Spi1*. *n*=4-6 per genotype from 5 independent litters.

liver (Tang et al., 2025; Qiu et al., 2024) consistently show that KCs express the highest levels of Igf1 among hepatic cell populations, establishing them as the predominant source of Igf1 during liver development. Together with our experimental models, these data identify KCs as a principal origin of Igf1 in the perinatal liver and reinforce their role as key regulators of early hepatic metabolism.

Our isotope-tracing experiments refine our understanding of this regulatory axis. Loss of KC-derived Igf1 did not alter baseline glycolytic or PDH fluxes, indicating that hepatocytes compensate for the absence of KC-derived Igf1 through alternative mechanisms. Despite this compensation, these livers responded to exogenous Igf1, which redirected glucose flux from glycolysis toward the TCA. This response was the opposite of the glycolytic shift seen after KC depletion and demonstrates that loss and restoration of Igf1 signaling exert reciprocal effects on hepatocyte metabolism. These findings suggest that KC-derived Igf1 fine-tunes the balance between glycolytic and oxidative metabolism rather than driving total carbon flux. They also reveal that developmental timing is crucial, since hepatocytes that develop without KCs undergo persistent metabolic reprogramming that cannot be fully corrected by later Igf1

exposure from KC-like cells. Nevertheless, we acknowledge that additional paracrine factors produced by KCs, such as Tnf (Diehl et al., 2020), as well as factors derived from other liver-resident cells, cooperate with Igf1 to orchestrate hepatocyte maturation and energy metabolism. This possibility is supported by the milder phenotype observed in the *KO^Igf1* model compared to the *KO^Spi1* and *KO^Csf1r* models and represents an important direction for future studies.

It remains to be determined whether Igf1 exerts its effects on perinatal hepatocyte metabolism via monomeric Igf1r, Insr homodimers, or their heterodimers. The role of macrophage-derived Igf1 in tissue development and function, also observed in the brain and gut (Yan et al., 2022; Rusin et al., 2024), underscores the specialized supportive functions of yolk sac-derived macrophages in their surrounding niche, particularly before growth factor production by other tissue cells commences.

Notably, this study shows that KC-like cells derived from sources other than the yolk sac or yolk sac-derived cells that establish a KC-like identity after E14.5 fail to fully support hepatocyte metabolism. We have previously proposed the concept of developmental programming of macrophages, suggesting that

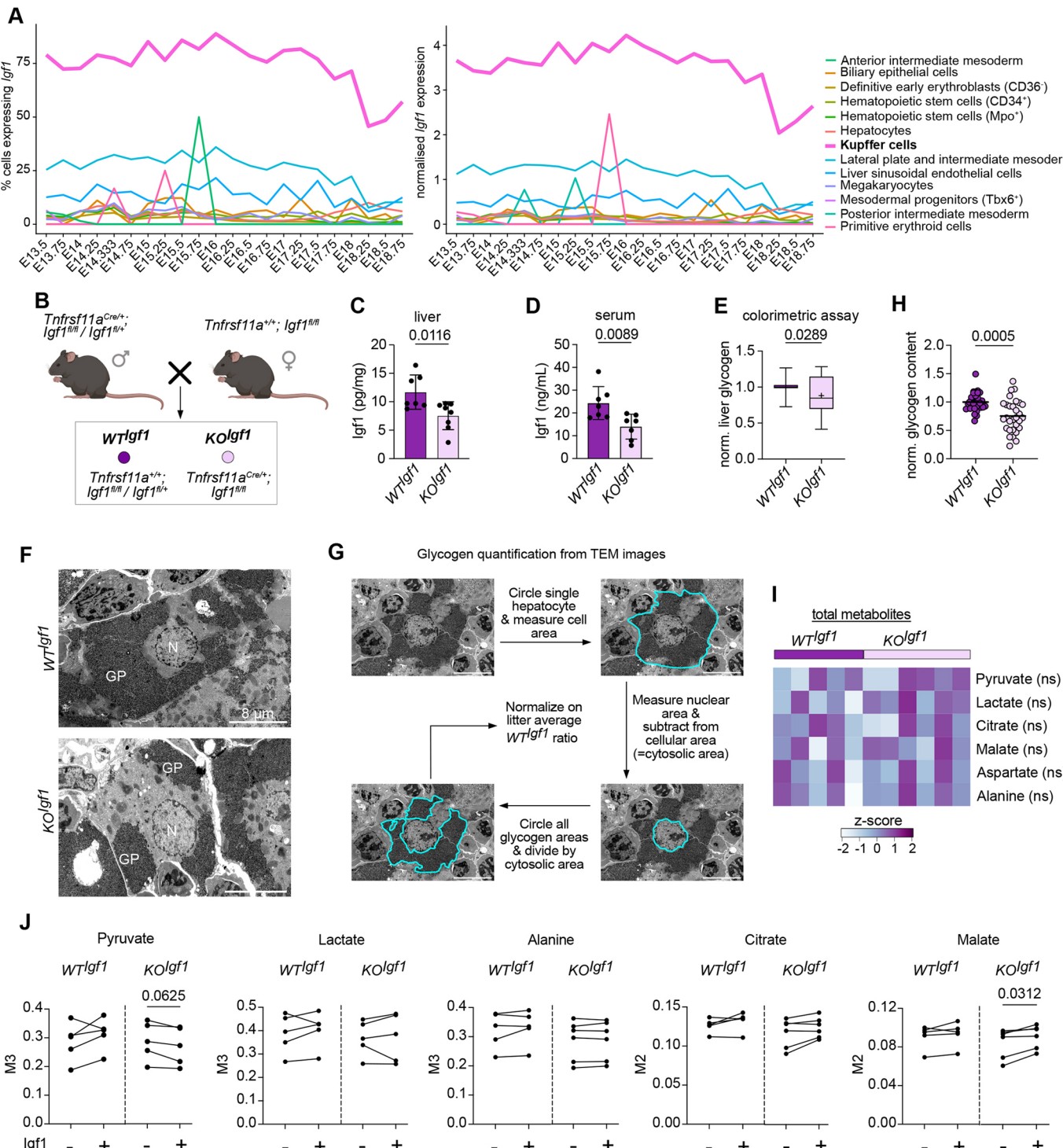

**Fig. 6. KC-derived Igf1 regulates glycogen homeostasis in hepatocytes at birth.** (A) Percentage of (left) and normalized (right) *Igf1* expression in the respective hepatic cell type during embryogenesis. (B) Breeding scheme to produce *KO^Igf1* mice and littermate controls (*WT^Igf1*). Created in BioRender by Mass, E., 2025. https://BioRender.com/jvsfc8p. This figure was sublicensed under CC-BY 4.0 terms. (C,D) Igf1 levels measured by ELISA on whole liver lysate (C) and serum (D) of *WT^Igf1* and *KO^Igf1* at P0. *n*=7-8 per genotype from 4 independent litters. Bar plots presented as mean±s.d. Unpaired Student's *t*-test. (E) Glycogen levels measured on whole liver lysates of *WT^Igf1* and *KO^Igf1* at P0. *n*=11-16 per genotype from 7 independent litters. Values were normalized per litter. The whiskers represent the 5-95% percentile, the box extends from the 25th to 75th percentiles and the horizontal line represents the median. Cross indicates the mean. Mann–Whitney test. (F) Representative transmission electron micrograph from *WT^Igf1*and *KO^Igf1* livers at P0. *n*=3-4 per genotype from 2 independent litters. GP, glycogen particle; N, nucleus. Scale bars: 8 µm. (G) Scheme indicating the quantification process of glycogen content in hepatocytes. (H) Hepatocyte glycogen content of *KO^Igf1* normalized to *WT^Igf1* littermates; each value represents one hepatocyte (ten hepatocytes were assessed per liver). *n*=3-4 per genotype from 2 independent litters. Mann–Whitney test. (I) Normalized total metabolite abundance in *WT^Igf1* and *KO^Igf1* livers following [U-$^{13}$C$_6$]-glucose tracing at P0. *n*=5-6 per genotype from 2 independent litters. Unpaired Student's *t*-test. ns, not significant (*P*>0.05). (J) Fractional enrichment of labeled metabolites following [U-$^{13}$C$_6$]-glucose tracing at P0 with and without the addition of exogenous Igf1 protein. Liver samples with and without Igf1 from the same animal are connected with a line. *n*=5-6 per genotype from 2 independent litters. Wilcoxon test.

their co-development with niche cells, ontogeny and longevity are crucial factors controlling their effector functions and, consequently, organ development and function (Mass and Gentek, 2021; Mass et al., 2023; Viola et al., 2024). Our findings reinforce that proper neonatal hepatocyte metabolism requires continuous interaction with yolk sac-derived KCs from the onset of liver organogenesis, a critical developmental window during which macrophage-derived cues shape hepatocyte maturation and long-term liver function.

## MATERIALS AND METHODS

### Mice

All investigations involving mice have been locally approved, and performed procedures have followed the guidelines from Directive 2010/63/EU of the European Parliament on protecting animals used for scientific purposes. The experiments were carried out following the German law of animal protection and met the criteria defined by the local institutional animal care committee [Landesamt für Natur, Umwelt und Verbraucherschutz (LANUV), North Rhine-Westphalia, Az 81-02.04.2018.A056 & Az 81-02.04.2023.A144]. The mice were housed under specific pathogen-free conditions, a 12-h light/dark cycle, with food and water available *ad libitum*. Timed matings were performed overnight. Females were plug checked the next morning, and pregnant females were monitored regularly. All mice were maintained on a C57BL/6JRcc background. To generate mice lacking macrophages during embryonic development $Tnfrsf11a^{Cre}$; $Spi1^{fl/fl}$ mice $Tnfrsf11a^{Cre}$; $Spi1^{fl/+}$ males were crossed with $Spi1^{fl/f}$ females (a similar scheme was employed for $Tnfrsf11a^{Cre}$; $Csf1r^{fl/fl}$ mice). For fate-mapping incoming myeloid cells $Tnfrsf11a^{Cre}$; $Spi1^{fl/fl}$ mice were crossed to $Ms4a3^{FlpO}$; $Rosa26^{FLF-tdTomato/FLF-tdTomato}$ mice ($Rosa26^{FLF-tdTomato}$ mice were originally obtained from The Jackson Laboratory, stock #032864). For prenatal time points, pregnant females were euthanized by cervical dislocation and the embryos were decapitated directly after cesarean section. All postnatal mice were separated from their mothers and decapitated before organ collection. All mice were genotyped referring to protocols and primers provided by The Jackson Laboratory or donating researchers. The sex of embryos and pups was not determined.

### Preparation of single-cell suspension for flow cytometry

Pre- and postnatal mice were euthanized as described, and the liver was removed and weighed. For E12.5 and E14.5, the whole liver was processed, and for P0 half was used for further steps. The tissue was cut into small pieces (or dissociated with a pipette for E12.5-E14.5), and incubated with 500 µl of a digestion mix (80 U/ml DNase I; Sigma-Aldrich, DN25) and 1 mg/ml collagenase D (Roche, 11088858001) in FACS buffer (0.5% bovine serum albumin, 2 mmol EDTA in PBS) for 30 min at 37°C. All following steps were performed on ice. The suspension was further passed through a 70 µm filter, and 5 ml cold FACS buffer was added. The samples were centrifuged at 400 $g$ for 5 min at 4°C and the supernatant was discarded. Red blood cell lysis was performed only for the P0 samples by dissolving the cells gently in 1 ml of cold RBC lysis buffer (155 mM $NH_4Cl$, 12 mM $NaHCO_3$ and 0.1 mM EDTA) and 5 min incubation. After adding 5 ml cold FACS buffer, the samples were again centrifuged (400 $g$, 5 min, 4°C). The supernatant was discarded, 50 µl of Fc-blocking buffer [anti-CD16/32 and 2% rat serum (liver) in FACS buffer] was added and the cells were gently resuspended. After 10 min incubation, all samples were filled up to 250 µl each and counted using a Guava cell-counting device. After pelleting, the cells were stained for 30 min (both primary and secondary antibody steps). The complete list of antibodies is supplied in Table S5. Samples were acquired with FACSymphony™ A5 (BD Biosciences) and analyzed using FlowJo™ Software.

### Mitochondrial staining

To evaluate mitochondrial status, liver cells were stained using MitoTracker® Green FM (Thermo Fisher Scientific, M7514) and MitoSOX™ Red (Thermo Fisher Scientific, M36008). Following cell counting, $5 \times 10^5$ cells were resuspended in FACS buffer and transferred into a 96-well plate. For stained samples, the volume was adjusted to 50 µl with

FACS buffer, while unstained controls were adjusted to 100 µl. Next, 50 µl of prewarmed (37°C) MitoTracker Green (0.25 µM) and MitoSOX Red (1 µM) solution in HBSS was added to the stained samples. The samples were incubated for 30 min at 37°C, after which 100 µl of FACS buffer was added to all wells. The plate was centrifuged at 400 $g$ for 5 min at 4°C, and the supernatant was carefully discarded. The cells were then incubated with the respective antibody mix (Table S5) for 30 min at 4°C. Following staining, the samples were immediately analyzed on the flow cytometer.

### Metabolic flow cytometry assay

Single-cell suspensions of P0 livers were blocked with anti-CD16/32 (1%) and rat serum (2%) in FACS buffer, followed by surface marker staining for 30 min at 4°C. Cells were washed and fixed in 1% paraformaldehyde for 5 min, then permeabilized for 15 min with PBS supplemented with 0.4% Triton™ X-100 (Sigma-Aldrich, X100). Intracellular staining was performed for 1 h, 4°C. Intracellular antibodies were conjugated in-house using lightning-link conjugation kits (Abcam) as previously described (Heieis et al., 2023). The complete list of antibodies is supplied in Table S5. Cells were analyzed on an ID7000™ 7-laser Spectral Cell Analyzer (Sony Cooperation).

### Liver glycogen assay

To access the glycogen content of perinatal liver tissue the Liver Glycogen Assay Kit from Abcam (ab65620) was used. The kit components were stored and dissolved following the manufacturer's instructions. The used liver samples were snap-frozen in liquid nitrogen and stored at −80°C until the experiment. After defrosting, 10 mg of liver tissue per sample were weighed in and the serum samples were diluted (1:25-1:100). The liver tissue was then washed shortly with cold PBS and put into 500 µl of ddH$_2$O. Homogenization was performed with a pestle on ice and the homogenates were boiled at 95°C for 10 min to inactivate enzymes. Further, the boiled homogenates were centrifuged at 4°C, maximal speed, and the supernatant was collected and stored at −20°C until the assay was performed. The main assay procedure was performed as described in the manufacturer's instructions. All samples and the glycogen standard curve were measured in technical duplicate. The plate was measure immediately after the last step with a Tecan Infinite M200 microplate reader at Ex/Em=535/587 nm. For analysis, the measured fluorescence from each sample was corrected with a glucose control (no addition of glucoamylase) to account for intrinsic differences in hepatic/systemic glucose levels.

### Igf1 ELISA

To determine the amount of Igf1 present in the perinatal liver and serum the Quantitative ELISA Mouse/Rat Igf1 Kit Liver Glycogen Assay Kit from R&D Systems (MG100) was used. The kit components were stored and dissolved following the manufacturer's instructions. The samples were snap-frozen in liquid nitrogen and stored at −80°C until the experiment was performed. On the day of the experiment, 10 mg liver tissue was homogenized in 500 µl ddH$_2$O on ice and cell membranes broken through two repeated freeze-thaw cycles. The liver homogenates were centrifuged at 5000 $g$ (5 min, 4°C) and the supernatant was collected. The serum samples were diluted 1:100 before analysis. For the ELISA, the manufacturer's instructions were followed. The optical density was determined using a Tecan Infinite M200 plate reader at 450 nm with correction at 540 nm.

### Insulin and glucagon ELISA

Serum samples were snap-frozen in liquid nitrogen and stored at −80°C until the experiment was performed. Serum insulin and glucagon concentrations were measured using the Mouse Insulin ELISA Kit (EMINS, Thermo Fisher Scientific) and Glucagon ELISA (Mercodia) according to the manufacturer's instructions.

### Lipidomics analysis

To evaluate differences in hepatic lipid metabolism, tandem mass spectrometry of extracted lipids was performed. For this purpose, 10 mg liver tissue was homogenized in 500 µl ddH$_2$O on ice. Then, 50 µl of the homogenate was transferred into a fresh Eppendorf tube and 500 µl Extraction Mix [CHCl$_3$/methanol 1/5 containing the following internal

standards: 210 pmol PE(31:1), 396 pmol PC(31:1), 98 pmol PS(31:1), 84 pmol PI(34:0), 56 pmol PA(31:1), 51 pmol PG (28:0), 28 pmol CL(56:0), 39 pmol LPA (17:0), 35 pmol LPC(17:1), 38 pmol LPE (17:0), 32 pmol Cer(17:0), 99 pmol, SM(17:0), 55 pmol GlcCer(12:0), 14 pmol GM3 (18:0-D3), 339 pmol TG(50:1-d4), 111 pmol, CE(17:1), 64 pmol DG(31:1), 103 pmol MG(17:1), 724 pmol Chol(d6) and 45 pmol Car(15:0)] was added. After 2 min of sonication in a bath sonicator, the samples were spun at 20,000 $g$ for 2 min. The supernatant was collected in a new Eppendorf tube and 200 μl chloroform and 750 μl of 1 M $NH_4Ac$ in $ddH_2O$ were added. Following quick manual shaking, the samples were centrifuged at 20,000 $g$ for 2 min again. The upper phase was carefully removed, and the lower phase was transferred into a new Eppendorf tube. The solvent was evaporated using a SpeedVac Vacuum Concentrator at 45°C for 20 min. The dried lipids were dissolved in 500 μl Spray Buffer [isopropanol, methanol, $ddH_2O$ (all MS grade), 10 mM ammonium acetate, 0.1% acetic acid] by sonication for 5 min. Until measurement with a Thermo Q Exactive™ Plus (Thermo Scientific) using positive mode, the samples were stored at −20°C. Before the acquisition, the samples were sonicated for 5 min. For downstream analysis, the raw spectral data was converted to .mzml files and loaded into the LipidXplorer software (Herzog et al., 2012). Using both the sample lipid and the previously added internal standard, the software calibrates the mass spectra and also discriminates based on the mass of the different lipid species. The sample lipid concentration in pmol was calculated referring to the intensity of the internal standard peak for each sample and its known concentration in the added internal standard. Samples with a high deviation of the internal standard were excluded from analysis. The overall abundance of the respective lipid class was obtained by summarizing the species' amounts. Lipidomics datasets are accessible via Metabolomics Workbench (Sud et al., 2016) under the study IDs ST003614 ($WT^{Spi1}/KO^{Spi1}$) and ST003702 ($WT^{Csf1r}/KO^{Csf1r}$).

### Phosphoproteome analysis
Liver tissues were lysed in 4% sodium deoxycholate and 100mM Tris-HCl (pH 8.5), followed by immediate boiling at 95°C for 5 min. The lysates were then sonicated for 20 min at 4°C. Protein concentrations were determined using the BCA protein assay, and samples were adjusted to 1 mg of total protein. Samples were reduced and alkylated using 10 mM tris(2-carboxyethyl)phosphine and 40 mM 2-chloroacetamide. Digestion was performed overnight at 37°C using lysC and trypsin at an enzyme-to-protein ratio of 1:100 (wt/wt). The digested peptides were treated with isopropanol (final concentration 50%), trifluoroacetic acid (TFA; final concentration 6%) and monopotassium phosphate ($KH_2PO_4$; final concentration 1 mM). Samples were mixed at 1500 rpm for 30 s, cleared by centrifugation (2000 $g$, 15 min), and the supernatants were incubated with $TiO_2$ beads (Titansphere Phos-TiO Bulk, GL Sciences) at a bead-to-protein ratio of 12:1 (wt/wt) for 5 min at 40°C. The beads were washed five times with 60% isopropanol and 5% TFA, and phosphopeptides were eluted using 40% acetonitrile (ACN) and 15% ammonium hydroxide ($NH_4OH$, 25%, HPLC grade). The eluates were collected by centrifugation (2000 $g$, 15 min) into clean PCR tubes and concentrated using a SpeedVac for 20 min at 45°C. Finally, phosphopeptides were desalted on SDB-RPS StageTips and resuspended in 7 μl of 2% ACN and 0.3% TFA for LC-MS/MS analysis.

Phosphopeptide samples were analyzed using an EASY-nLC 1000 HPLC system (Thermo Fisher Scientific) coupled to a Q Exactive HFX mass spectrometer (Thermo Fisher Scientific) via a nano-electrospray ion source. The samples were loaded onto a 50-cm column with a 75 μm inner diameter, packed in-house with C18 1.9 μM ReproSil particles (Dr. Maisch GmbH). The column temperature was maintained at 50°C using a custom-built column oven. Phosphopeptides were separated using a buffer system consisting of buffer A (0.1% formic acid) and buffer B (0.1% formic acid, 80% ACN). A 120-min gradient was used for elution, starting at 5% buffer B and increasing stepwise to 30% in 95 min, 60% in 5 min, and 95% in 20 min at a flow rate of 300 nl/min. Phosphopeptides were analyzed using a data-independent acquisition MS method. This included one full scan with a range of 300-1650 m/z (AGC target=$3e^6$, R=120,000 at 200 m/z, maximum injection time=60 ms) followed by MS/MS scans with 32 windows where precursor ions were fragmented with higher-energy collisional dissociation (nce=27%, AGC target=$1e^6$, R=30,000 at 200 m/z, maximum injection

time=54 ms). Data acquisition was performed using Xcalibur (v4.4.16.14, Thermo Fisher Scientific).

Raw MS data files were processed using Spectronaut software (v14.3.200701.47784). Mass spectra were searched against the mouse UniProt FASTA database (July 2019, 63,439 entries) with a false discovery rate (FDR) of <1% at the protein and peptide levels. Database search parameters allowed a minimum peptide length of seven amino acids and a maximum of two missed cleavages. Variable modifications included phosphorylation on serine, threonine and tyrosine residues, N-terminal acetylation, and methionine oxidation, while cysteine carbamidomethylation was set as a fixed modification. All bioinformatics analyses were conducted in R (v4.0.4). Quantified phosphosites were filtered to include only those with at least three valid values across biological replicates in at least one condition. Missing values were imputed using a Gaussian distribution with a width of 0.3 and a downshift of 1.8 standard deviations of the measured values. Differentially regulated phosphosites were identified using an unpaired, two-sample t-test Student's t-test with a permutation-based FDR of 5%. A 1D annotation enrichment analysis was applied to identify systematic enrichment or depletion of annotations and pathways (FDR<5%).

### PAS staining
Periodic acid-Schiff's with Hematoxylin (PAS-H) staining was performed on paraffin-embedded tissue sections (5 μm thick) to analyze glycogen storage in neonatal liver tissues. Tissue sections were incubated at 65°C for 30 min to melt the paraffin. Rehydration was performed by passing the sections through a series of steps, each for 3 min: two changes of xylene, followed by 100%, 95%, 90%, 80% and 70% ethanol, and finally distilled water. Rehydrated sections were incubated in periodic acid solution (Carl Roth, HP00) for 10 min to oxidize glycogen, rinsed in tap water for 3 min, and briefly rinsed in distilled water. The sections were placed in Schiff's reagent (Carl Roth, X900.2) for 15 min. Following staining, sections were washed in running tap water for 10 min then briefly rinsed in distilled water. Counterstaining was performed using Hematoxylin (Carl Roth, T865) for 45 s to visualize nuclei. Excess Hematoxylin was removed by washing in tap water for 3 min, followed by a brief rinse in distilled water. Sections were passed through distilled water, 70%, 80%, 90%, 95% and 100% ethanol, each for 3 min, and then cleared in two changes of xylene. Sections were mounted with Entellan (Merck, 107961) and covered with glass coverslips. Stained sections were examined and recorded with a light microscope. For analysis of acquired images Fiji analysis software (v2.1.0/1.53c) was used. First, the color threshold was set for the images and they were converted to 8-bit images. After setting an auto threshold, they were reverted, and the color deconvoluted. Signal intensity for the PAS image plane was measured, and all values were normalized using the respective WT reference.

### Transmission electron microscopy
Transmission electron microscopy of mouse liver samples from the $KO^{Spi1}$ mouse model was performed as described previously for other organs (Fazio et al., 2022; Welz et al., 2022). To quantify cristae area of mitochondria, images were acquired at a nominal magnification of ×20,000 and mitochondria were manually singled out using ImageJ. Subsequently, we applied an in-house-developed script utilizing ImageMagick (www.imagemagick.org). This script binarized the mitochondrial images applying the same threshold value for all samples. We then counted the black pixels in each mitochondrion as a measure for mitochondrial mass. To obtain the cristae area, we subtracted the black pixels from the overall mitochondrial pixel number. To calculate the cristae area per mitochondria, we divided the cristae area through the overall area. Livers obtained from P0 $KO^{Igf1}$ mice and littermate controls were fixed overnight in 0.1 M Caco (sodium cacodylate) buffer with 4% paraformaldehyde and 2% glutaraldehyde. Subsequently, they were washed with PBS, embedded in agarose, and cut into 50 μm thin cuts using a vibratome. The sections were again fixed overnight as before and washed with 0.1 M Caco buffer. Until further processing, the cuts remained in 0.1 M Caco buffer at 4°C. Image analysis and quantification of cellular glycogen areas were performed in Fiji (ImageJ). Cell boundaries, nuclear area and glycogen area within a hepatocyte were manually outlined using the polygon selection tool, and the area was determined. The nuclear area was subtracted from total cellular area to obtain cytosolic area, which was in turn divided by the sum of

all glycogen-occupied areas. The resulting ratio was normalized to the average $WT^{Igf1}$ ratio of the respective litter. Ten hepatocytes were analyzed per mouse, and the average normalized values plotted.

### Perinatal liver culture and metabolic tracing

Newborn pups were euthanized immediately after birth by decapitation, and the liver was collected in ice-cold culture buffer (Williams Medium supplemented with 10% fetal calf serum, 1% L-glutamine and 1% Penicillin-Streptomycin). For each replicate, 2-3 mg of the largest liver lobe was carefully dissected and placed into a 48-well plate containing ice-cold culture buffer. All subsequent steps were carried out in a sterile hood to maintain sterility. Each liver piece was transferred onto a porous cell culture insert (0.4 µm pore size), which was placed in a well of a 24-well plate containing 400 µl prewarmed culture buffer. The plate was incubated at 37°C with 5% $CO_2$ for 2 h. This incubation period allowed for the simultaneous genotyping of the pups to ensure that only $WT^{Spi1}$ and $KO^{Spi1}$ samples were included in the tracing experiment, excluding heterozygous ($HET^{Spi1}$) samples. After genotyping, the inserts with the liver tissue pieces were transferred to a new 24-well plate containing 400 µl prewarmed tracer medium (Williams Medium without glucose, supplemented with 10% fetal calf serum, 1% L-glutamine and 25 mM $C^{13}$-labeled glucose). Recombinant Igf1 protein (R&D Systems, 791-MG) was added in the same way at a final concentration of 100 ng/ml. The samples were incubated for 5 h at 37°C with 5% $CO_2$. Following the incubation, 100 µl of medium was collected and snap-frozen. The liver tissue pieces were briefly washed with 0.9% NaCl and then placed in 250 µl of methanol pre-cooled to −20°C. To each sample, 250 µl of pre-cooled MS-grade water containing 1 µg/ml D6-glutaric acid (used as an internal standard) was added, and the tissue was homogenized. The homogenate was transferred to a fresh tube and stored at −80°C until further processing. Metabolites were derivatized using a Gerstel MPS with 15 µl of 2% (w/v) methoxyamine hydrochloride (Thermo Scientific) in pyridine and 15 µl N-tertbutyldimethylsilyl-N-methyltrifluoroacetamide (MTBSTFA) with 1% tert-butyldimethylchlorosilane (Regis Technologies). Derivatives were measured by GC/MS with a 30 m DB-35MS+5 m Duraguard capillary column (0.25 mm inner diameter, 0.25 µm film thickness) equipped with an Agilent 7890B gas chromatograph (GC) connected to an Agilent 5977A mass spectrometer (MS). The GC oven temperature was held at 80°C for 6 min and subsequently increased at 6°C per minute until reaching 280°C where it was held for 10 min. The quadropole was held at 150°C. The MS source operated under electron impact ionization mode at 70 eV and was held at 230°C.

Full scan (70-800 m/z, 3.9 scans per second) as well as targeted ion chromatogram measurements were conducted for pyruvate (174, 175, 176, 177, 178, 179; ten scans per second), lactate (261, 262, 263, 264, 265, 266, 267; ten scans per second), alanine (260, 261, 262, 263, 264, 265; ten scans per second), citrate (591, 592, 593, 594, 595, 596, 597, 598, 599, 600; ten scans per second), malate (419, 420, 421, 422, 423, 424, 425, 426; ten scans per second) and aspartate (418, 419, 420, 421; ten scans per second). All chromatograms were analyzed with MetaboliteDetector (Hiller et al., 2009). [U-$^{13}C_6$]-glucose tracing datasets are accessible via Metabolomics Workbench (Sud et al., 2016) under the project number ST003615.

### Bulk RNA sequencing

Cells were stored in TRIzol at −80°C. The cDNA library for sequencing was prepared following the SMART-Seq2 protocol. mRNA was isolated, primed using poly-T oligonucleotides, and then converted into cDNA via SMART reverse transcription. Pre-amplification was performed using SMART ISPCR, followed by fragmentation with the Nextera XT DNA Library Preparation kit (Illumina), amplification and indexing. Library fragments were then selected by size (300-400 bp) and purified using SPRIBeads (Beckman-Coulter). Size distribution of the cDNA libraries was analyzed using the Agilent high-sensitivity D5000 assay on the Tapestation 4200 system (Agilent). Quantification of cDNA libraries was completed with a Qubit high-sensitivity dsDNA assay (Thermo Fisher). Sequencing employed a 75 bp single-end configuration on the NextSeq500 system (Illumina), using the NextSeq 500/550 High Output Kit v2.5. Kallisto pseudo-alignment was applied to quantify transcript abundances from bulk RNA-seq data (Bray et al., 2016). The files were processed using Kallisto for transcript quantification, with the Gencode M16 mouse annotation

(https://www.gencodegenes.org/mouse/release_M16.html) applied to adjust for library size based on the average transcript length. Read counts were normalized with DESeq2 default size factor estimation, which corrects for sequencing depth across samples. Low-expressed genes, defined as those with fewer than ten total counts, were removed to enhance signal detection. Statistical significance in gene expression was determined using the Wald test for pairwise comparisons (default method in DESeq2) Adjusted $P$-values were calculated using the Benjamini–Hochberg method to control the FDR. Genes with an adjusted $P$-value<0.05 and |log2fold change|>1.3 were considered significantly differentially expressed. Normalized count values for individual genes were visualized to assess expression differences. Genes were ranked by differential expression for each condition, and the resulting ranked gene list was used for GSEA on selected gene sets.

### scRNA-seq library preparation and analysis

Single-cell suspensions were prepared as previously described. The cells were then spun for 3 min at 50 $g$ to enrich for hepatocytes. Both hepatocytes and liver immune cells were utilized for single-cell analysis. To ensure that hepatocytes were appropriately sized to fit into the wells of the Seq-Well array, the cells were fixed to induce controlled shrinkage. Hepatocytes were resuspended in PBS and fixed by the slow, dropwise addition of ice-cold methanol at a 1:5 ratio (PBS:methanol) while gently stirring the solution between additions. The suspension was incubated at −20°C for 30 min, followed by 5 min on ice. Subsequently, the fixed cells were centrifuged at 1000 $g$ for 5 min at 4°C, and the supernatant was carefully discarded. The hepatocytes were resuspended in 500 µl of rehydration buffer containing 3×SSC (Sigma-Aldrich, S6639-1L), 1 mM dithiothreitol (Thermo Fisher Scientific, R0862), 0.2 U/µl RNase inhibitor (Thermo Fisher Scientific, AM2696) and 2 mM flavopiridol (Sigma-Aldrich, F3055). Cell loading, barcoding and library preparation primarily followed the Seq-Well S3 protocol (Hughes et al., 2020), with two arrays per sample. Seq-Well arrays were set up as described by Gierahn et al. (2017). Each array was loaded with approximately 110,000 barcoded mRNA capture beads (ChemGenes, MACOSKO-2011-10) and 30,000 cells. Following cell loading, cells were lysed, mRNA captured, and cDNA synthesis was performed. For whole-transcriptome amplification, beads from each array were divided into 18-24 PCR reactions with approximately 3000 beads per reaction (95°C for 3 min, four cycles of 98°C for 20 s, 65°C for 45 s, 63°C for 30 s, 72°C for 1 min; followed by 16 cycles of 98°C for 20 s, 67°C for 45 s, 72°C for 3 min; final extension at 72°C for 5 min) using the KAPA HiFi Hotstart Readymix PCR Kit (Kapa Biosystems, KK2602) and SMART PCR Primer (AAGCAGTGGTATCAACGCAGAGT). Pooled PCR reactions (six to eight per pool) were purified using AMPure XP SPRI Reagent (Beckman Coulter) with sequential 0.6× and 1× volumetric ratios. For library tagmentation and indexing, 200 pg of DNA from each purified WTA pool was tagmented with the Nextera XT DNA Library Preparation Kit (8 min at 55°C), followed by Tn5 transposase neutralization (5 min at room temperature). Illumina indices were then attached to the tagmented products (72°C for 3 min, 98°C for 30 s; 16 cycles of 95°C for 10 s, 55°C for 30 s, 72°C for 1 min; final extension at 72°C for 5 min). The library products were purified using AMPure XP SPRI Reagent at 0.6× and 1× volumetric ratios. The final library quality was assessed using a High Sensitivity D5000 assay on a TapeStation 4200 (Agilent) and quantified with the Qubit high-sensitivity dsDNA assay (Invitrogen). Seq-Well libraries were pooled equimolarly and clustered at a 1.25 nM concentration with 10% PhiX on a NovaSeq6000 system (S2 flow cell, 100 bp v1.5 chemistry). Sequencing was paired-end, using a custom Drop-Seq Read 1 primer for 21 cycles, eight cycles for the i7 index, and 61 cycles for Read 2. Single-cell data were demultiplexed using bcl2fastq2 (v2.20). Fastq files from Seq-Well were processed in a snakemake-based pre-processing pipeline (v0.31, available at https://github.com/Hoohm/dropSeqPipe) that utilizes Drop-seq tools provided by the McCarroll lab (Macosko; Basu et al., 2015). STAR alignment within the pipeline was performed using the murine GENCODE reference genome and transcriptome (mm10 release vM16; Team 2014).

To analyze liver cells, UMI-corrected expression matrices were processed in R using Seurat (v4.1.0) (Hafemeister and Satija, 2019) From an initial dataset of 407,803 barcodes and 43,126 genes, only protein-coding genes were retained, reducing the dataset to 16,131 genes. Quality control

measures (Luecken and Theis, 2019) identified 27,258 high-quality cells. Ambient RNA contamination was corrected using SoupX (Young and Behjati, 2020) further refining the dataset to 27,235 cells. Data normalization and scaling were performed with SCTransform (Hafemeister and Satija, 2019) and dimensionality reduction was achieved through principal component analysis. The first 21 principal components were selected for downstream analyses, including uniform manifold approximation and projection (UMAP) visualization and the construction of a shared nearest neighbor graph. Clustering was performed using the Louvain algorithm (Traag et al., 2019), with resolution determined by Clustree v0.4.3 (Zappia and Oshlack, 2018) and NbClust v3.0 (Charrad et al., 2014) analyses. Marker genes for each cluster were identified using a Wilcoxon rank sum test (Haynes, 2013), applying a $\log_2$ fold-change threshold of 0.25 and requiring expression in at least 10% of cells. Cluster annotation was based on canonical marker genes. To investigate functional phenotypes in hepatocytes, a Wilcoxon rank sum test was performed without thresholds to identify all DEGs between $WT^{Spi1}$ and $KO^{Spi}$ genes were ranked by average $\log_2$ fold-change for each genotype and the top 100 genes were used for GO enrichment analysis (Yu et al., 2012; Haynes, 2013) and the Mouse annotation database (org.Mm.eg.db) v3.12.0. GSEA was conducted using ClusterProfiler to identify enriched biological processes from the GO database (Ashburner et al., 2000). Statistical significance of enriched terms was evaluated using a hypergeometric test (Yu et al., 2015) with Bonferroni correction for multiple comparisons (Yu et al., 2012, 2015; Haynes, 2013). Terms with adjusted $P$-values <0.05 were considered significantly enriched. Developmental trajectories of hepatocyte subclusters were inferred using Monocle3 v1.0.0 (Cao et al., 2019). For trajectory construction, hepatocytes were subsetted from the liver scRNA-seq dataset and the first 30 dimensions of the scRNA-seq data were utilized to generate a UMAP focused exclusively on hepatocytes. A connection matrix was computed using a modified partitioned approximate graph abstraction (PAGA) algorithm (Wolf et al., 2017), followed by significance testing of connections between clusters identified through PAGA and Louvain clustering. Subsequently, a principal graph was constructed in the low-dimensional space using a modified SimplePPT algorithm (Mao et al., 2015). Pseudotime progression was determined by designating the node within most immature hepatocyte cluster 0 as the root node based on hepatocyte maturation marker expression, enabling the inference of developmental relationships among hepatocyte subclusters. Intercellular communication between hepatocytes and KCs was analyzed following the methods described by Jin et al. (2021), utilizing CellChat v1.1.3. The Seurat object was split into three datasets based on genotypes. Each dataset was converted into a CellChat object using normalized and log1p-transformed expression matrices (Jin et al., 2021). The dataset was then subset to include only genes associated with known signaling pathways. Overexpressed ligand and receptor genes were identified using the Wilcoxon rank sum test, without applying a $P$-value threshold. Noise was mitigated by calculating the average expression of each gene for each cell group, with weighted summation of gene expression quantiles. The prediction of gene–gene interactions relied on protein–protein interaction (PPI) networks sourced from STRINGDb (Szklarczyk et al., 2019) under the assumption that physical interactions between ligands and receptors follow the law of mass action. To achieve this, signaling gene expression profiles were projected onto the PPI network using random walk network propagation (Cowen et al., 2017). The interaction probability (or strength) was modeled using the projected data, applying a trimming fraction of 0.01. Communication pathways between cell groups were identified through permutation testing. Additionally, social network analysis tools from the sna package (Butts, 2008) were used to calculate information flow metrics, including out-degree, in-degree, flow betweenness, and information centrality, to evaluate intercellular communication.

## Code and data availability

All steps, including cleaning, dimensionality reduction, clustering, DEG testing, and GO enrichment analysis, were conducted using the docker image alefrol94/scrnaseq.analysis:reticulate. Additional packages were managed and tracked using the renv package (v0.14.0). Trajectory analysis of hepatocyte subsets, performed with Monocle3 (Cao et al., 2019), utilized the docker image jsschrepping/r_docker:jss_R403_S4cran. The complete analysis code can be found at https://github.com/alefrol638?tab=repositories.

scRNA-seq datasets are accessible via NCBI's Gene Expression Omnibus (GEO) repository under accession number GSE285047. The bulk RNA-seq raw transcriptome files and count data have been deposited in GEO and are accessible under accession number GSE283799. Igf1 expression across development was visualized based on data reported by Qiu et al. (2024).

## Experimental design, quantification and statistical analysis

For the design of experiments, scientists were unaware of the experimental groups whenever possible, as the animals exhibited no overt phenotype at the time of sample processing. This was applied during various stages, such as histological and transmission electron microscopy (TEM) analyses, where subjective interpretation could introduce bias, but it was not feasible in all assays. For example, genotyping PCR and certain analyses, such as flow cytometry, revealed clear phenotypic differences, which could not be masked from experimenters. To account for variability between experimental runs, we normalized data to the mean value(s) of littermate controls collected on the same day. This normalization was particularly important for measurements such as the mean fluorescence intensity of flow cytometry markers and most metabolic analyses, ensuring consistency and comparability across experiments. Every reported sample ($n$) value represents the number of biologically independent replicates. No statistical methods were used to predetermine sample sizes, but sample numbers were based on standards in the field and experimental feasibility. All statistical analyses, except for those related to sequencing data, were conducted using GraphPad Prism software (v5-8; GraphPad Software, RRID:SCR_002798). Mann–Whitney $U$-tests and paired/unpaired Student's $t$-tests (e.g. paired tests for histological analysis due to variability in staining intensity across experiments) were used for comparisons between two groups, and mixed-effects model (REML) for comparisons between four groups were performed depending on data distribution and experimental design. Wald test was performed on the fold-change gene expression of KCs and KC-like cells for pairwise comparisons. Differences were considered statistically significant at $P$<0.05. Each dataset is representative of at least two independent experiments, with a minimum of three animals per group. Detailed descriptive statistics, the specific statistical tests performed, and the number of samples analyzed are provided in the corresponding figure legends.

## Acknowledgements

We thank Cornelia Cygon for support in the lab. We thank Andrea Eichhorn, Dr Hannes Beckert, Dr Kristiano Ndoci and Elke Kretzschmar for their help with the TEM images. We thank Yasuhiro Kobayashi for providing $Tnfrsf11a^{Cre}$ mice and Florent Ginhoux for providing the $Ms4a3^{FlpO}$ mice. We would like to thank the Flow Cytometry Core Facility of the Mathematical and Natural Sciences Faculty and the Flow Cytometry Core Facility as well as the Microscopy Core Facility of the Medical Faculty at the University of Bonn for providing support and instrumentation funded by the Deutsche Forschungsgemeinschaft (DFG, German Research Foundation) - Project numbers 341039622, 144734146, 471514137, 01EO2107 (BMBF), 388171357, 553871989. The authors used ChatGPT (OpenAI) for language correction and proofreading. Thereafter, the authors reviewed and edited the content and take full responsibility for the content of the article. This work is supported by Metabolomics Workbench/National Metabolomics Data Repository (NMDR) (U2C-DK119886), Common Fund Data Ecosystem (CFDE) (3OT2OD030544) and Metabolomics Consortium Coordinating Center (M3C) (1U2C-DK119889).

## Competing interests

The authors declare no competing or financial interests.

## Author contributions

Conceptualization: E.M.; Data curation: A.F., B.B., F.N., J.C., P.A., M.B.; Formal analysis: N.M., D.J.H., I.S., B.B., F.N., J.C., P.A., L.B., E.M.; Funding acquisition: C.T., F.M., K.H., M.D.B., E.M.; Investigation: N.M., D.J.H., I.S., N.B.-S., M.F.V., M.Y.; Methodology: N.M., D.J.H., E.M.; Resources: C.T., F.M., K.H., M.D.B., E.M.; Software: A.F.; Supervision: C.T., F.M., K.H., M.D.B., E.M.; Validation: N.M., D.J.H.; Visualization: N.M., D.J.H., A.F., B.B., F.N., J.C., M.F.V., P.A., E.M.; Writing – original draft: N.M., D.J.H., A.F., E.M.; Writing – review & editing: N.M., D.J.H., I.S., B.B., F.N., J.C., N.B.-S., M.F.V., M.Y., P.A., L.B., M.B., C.T., F.M., K.H., M.D.B., E.M.

## Funding

This work was supported by the Deutsche Forschungsgemeinschaft (DFG; German Research Foundation) (under Germany's Excellence Strategy EXC2151-390873048 to E.M., M.D.B., F.M., C.T., L.B.; Project-ID 432325352 – SFB 1454 to

E.M., M.D.B., K.H., F.M., C.T.; GRK1873/2 to N.M., E.M.; the IRTG program GRK2168, Project number 272482170 to M.D.B.; FOR5547, Project-ID 503306912 to E.M.; FOR5775, Project-ID 533863915 to E.M.; ImmuDiet, Project ID 432325352 to L.B.), an EMBO Postdoctoral fellowship (873-2023 to M.F.V.) and the European Research Council (ERC) under the European Union's Horizon 2020 research and innovation program (851257 to E.M.; 101163024, POLIS, to L.B.). Open Access funding provided by the University of Bonn. Deposited in PMC for immediate release.

## Data and resource availability

The lipidomics datasets are available at the NIH Common Fund's National Metabolomics Data Repository (NMDR) website, the Metabolomics Workbench (https://www.metabolomicsworkbench.org; Sud et al., 2016) under Project IDs PR002234 ($WT^{Spi1}/KO^{Spi1}$) and PR002234 ($WT^{Csf1r}/KO^{Csf1r}$). The data can be accessed directly via the Project DOIs 10.21228/M8VJ9K ($WT^{Spi1}/KO^{Spi1}$) and 10.21228/M8VJ9K ($WT^{Csf1r}/KO^{Csf1r}$). [U-$^{13}C_6$]-glucose tracing datasets are accessible via Metabolomics Workbench under project ID PR002234 and can be accessed directly via the Project DOI 10.21228/M8VJ9K. scRNA-seq datasets are accessible via the GEO repository under accession number GSE285047. Bulk RNA-seq raw transcriptome files and count data have been deposited in GEO and are accessible under accession number GSE237408. All other relevant data and details of resources can be found within the article and its supplementary information.

## The people behind the papers

This article has an associated 'The people behind the papers' interview with some of the authors.

## Peer review history

The peer review history is available online at https://journals.biologists.com/dev/lookup/doi/10.1242/dev.204962.reviewer-comments.pdf

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
