## [Peer Review File · Development (Cambridge, England)]

Kupffer cells control neonatal hepatic metabolism via Igf1 signaling

Nikola Makdissi, Daria J. Hirschmann, Aleksej Frolov, Inaam Sado, Bastian Bennühr, Fabian Nikolka, Jingyuan Cheng, Nelli Blank-Stein, Maria Francesca Viola, Mohamed Yagmour, Philipp Arnold, Lorenzo Bonaguro, Matthias Becker, Christoph Thiele, Felix Meissner, Karsten Hiller, Marc D. Beyer and Elvira Mass
DOI: 10.1242/dev.204962

Editor: James M. Wells

Review timeline

Submission to Review Commons:	18 February 2025
Submission to Development:	20 May 2025
Editorial decision:	30 May 2025
First revision received:	3 November 2025
Accepted:	1 December 2025

Reviewer 1:

Evidence, reproducibility and clarity

****Summary:****

Kupffer cells (KCs), the resident liver macrophages, inhabit the sinusoids of the liver where they serve as immune cells, defending against blood-borne pathogens. Their closest cellular neighbor are LSECs (Liver Sinusoidal Endothelial Cells), but they also interact with hepatocytes and stellate cells. KCs originate from hematopoietic cells. Initial hematopoiesis takes place in the blood islands of the yolk sac, from where precursors enter the blood stream and colonize the liver to form KCs. Yolk-sac derived KCs remain in the liver throughout life as an independent population. However, upon depletion, the KC pool can be replenished from circulating monocytes.

In their present study, the authors aim to investigate the role of resident KCs in the neonate mouse liver. To this end they deplete the prenatal resident macrophage population through a tissue-specific knock-out (targeting all pre-macrophages) of *Spi1* or *Csf1r*. In these models the prenatal KC population is near-depleted and by birth replenished largely by local precursors, rather than the circulating pool. They then compare the metabolic profile of wild-type and KO mice, and find metabolic deficits in the KOSpi1 livers (lowered glycogen). In scRNA-seq comparing WT and KOSpi1 mice, *Igf1* and *Igfr1* interactions of WT and KOSpi1 hepatocyte/KCs are reduced. They conclude that *Igf1* production is performed by yolk-sac derived, resident KCs, and that this *Igf1* is required for regulated hepatocyte glycogen metabolism in neonate mice.

****Major comments:****

1. Mouse models: small contribution of monocyte-derived KC-like cells to the total KC pool
Only 5-7% of the KC population at P0 are derived by circulating monocytes, so the majority of impaired KC-like cells in KOSpi1 is still yolk-sac derived. Therefore, the claim that yolk-sac origin is critical is a bit misleading, since they apparently only in minority contribute to the re-population of the KC pool. If the authors wanted to keep their claim on the failure of the HSC-derived pool, they would need to separately characterize *Ms4a3*-positive KC-like cells in WT and KOSpi1, and compare them, demonstrating loss of marker expression in *Ms4a3*-positive KOSpi1 KCs. Probably it is more useful to rephrase that the replenishment of the KCs is mainly driven by

resident precursors, and therefore circulating replenishment is simply no option for restoration.
2. Marker gene expression levels (Fig 1K)

It would be important to see expression levels of important KC markers Clec2f, Clec4f, Crig in the main figure.

3. Replicate numbers, n numbers, time points

Please state the n numbers (biol. replicates or how often the experiment was repeated), and the time point on top of every plot, or at least in the beginning of a new experiment e.g. when scRNAseq data is first introduced (annotate N, Ncells, timepoint). This will improve readability a lot.

4. Igf1 expression reduced in KOSpi1 or Csf1r? (qPCR or WB)

It would be very important to show whether Igf1/Insr/Gcg levels differ in E14.5/E12.5 and P0KCs between WT/KOSpi1 or Csf1r to keep the Igf1 hypothesis in title and text. Maybe another cell population can be gated and compared as a control, to see whether we have cell-type specificity here. Additionally, the same expression profile would be expected in the already present RNA-Seq and scRNA-Seq data of KCs, and showing these as well would be needed to prove findings are consistent across methods.

5. Incomplete maturation of KOSpi1 KC-like cells may hinder proper monocyte recruitment

KOSpi1-KCs downregulate "genes such as Tnf, Il6, Il1a, Il1b, and Cxcl10 falling into the term 'inflammatory response' (Figure 1N)". However, we know that to open the niche and recruit monocytes, KCs have to release tumour necrosis factor (TNF) and IL-1 (Bonnardel, Immunity <https://doi.org/10.1016/j.immuni.2019.08.017>, 2019). May this indicate that insufficient restoration of KCs by circulating monocytes is potentially an artifact of the KOSpi1-KCs (and their reduced immune function)? This may be discussed in the text and statements on failed replenishment may be toned down.

6. Igf1 role - but what about glucagon levels?

The authors find lower liver glycogen levels at birth in KOCsf1r / KOSpi1 mice, and I agree to their hypothesis this being in connection with metabolic dysregulation. However, glucagon levels in KOs could just be higher at birth and thus lead to more liver glycogen being mobilized. The authors even mention glucagon in the introduction but seem not to measure it. Assessment of glucagon levels or other important players of glycogen metabolism in P0livers of WT vs KOSpi1/Csf1r would be an important control.

7. ScRNAseq replicate numbers

It is not clear how many biological replicates contributed to the single cell data. I don't see replicate integration in the methods, so I assume this is only one replicate which is probably not sufficient. Please provide more information and additional scRNA-seq replicates (usually 3).

8. Complex niche maturation deficit during embryonic development in Kos

Knockouts of Spi1 and Csf1r are not temporally restricted, thus the metabolic deficits observed in the knockout livers could likewise result from any interference in liver niches (KC/LSEC/Hepatocyte/Cholangiocyte/HSC interaction) throughout prenatal development. E.g., the depletion of KCs could affect LSECs and only through LSECs hepatocytes or other related cell types. A thorough characterization of all the other cell types (e.g. LSECs, Stellate cells,...) including cell-cell interactions with hepatocytes in KO/WT livers should show no deficits compared to WT cells. Again, it would be important to show Igf1 and Insr expression across all cell types. Importantly, cell-cell-interactions involving especially Igf1 and Igf1r between all available cell types should be shown to rule out other cell types contributing. Until then, the assumption that impaired KC-like cells are the central contributor to hepatocyte dysregulation is speculative. Alternatively, a temporally-controlled depletion of resident KC's Igf1 functionality at birth (once the niche has readily established) would be desirable, but probably more difficult.

9. Validation of cells contributing to hepatocyte Igfr activation

Once the previous analysis has been performed and hints at KCs being the major source of Igf1, a staining for Igf1 (or probing the transcript) displaying co-localization with KCs (labelled with specific antibody) would be important to validate scRNA-seq observations.

10. Igf1 KO / Fig 4G-J

In figure 4G the reduction if liver Igf1 is used to validate the KC-Igf1 KO, but actually it is not so dramatic, hinting at other Igf1 sources. The statement "We first confirmed that macrophages were the primary producers of Igf1 at P0." needs to be toned down urgently, looking at 4G I can immediately count at least 4 dots with exactly the same Igf1 level as in the WT. Again the glycogen (J) reduction looks very moderate. The strong statement on Igf1 role again may need to be toned down. Alternatively, a rescue experiment restoring hepatocyte transcriptional/metabolic deficits and glycogen storage upon Igf1 delivery may be provided.

11. Title/Scope

The metabolic and transcriptomic characterization of neonate KOSpi1 liver is interesting and thorough. Based on the findings, the association with the Igf1 phenotype without rescue experiments is less strong. Maybe the title can be toned down accordingly, e.g. stating "Characterization of hepatic metabolism and Igf1 signaling in two prenatal KC depletion mouse models"

****Minor comments:****

1. Quantification plots (e.g. 1C,G,H,J,L...) would benefit from adding the time point in the title. Please add time points for all figures to help readability.
2. Some of the references are not formatted correctly
3. The reduction in PAS signal in Fig 2 doesn't look very convincing to me, also it is not significant (p0.07). Please change the text accordingly.
4. There is two Figure 4
5. Fig.4 G/H says Igf1 KO, but in the legend WTSp1 and KOSpi1 at P0. For interpretation this is important, please also check the other legends.

****Referees cross-commenting****

I appreciate my co-reviewer's comments.

Their suggestion to focus on spi1KO makes sense. Still I would suggest keeping the information on csf1r in the supplements. They are indeed different at birth, this information may be valuable for some readers, additionally more models may make the paper more likely to be cited.

I completely agree to their comment on the Ms4a3, I see this related to my comments #1 and #5.

Their comment on Tim4 makes sense, I see this in line with my request for more KC markers #2, especially Clec4f. I would add that these cells are still only 5% HSC derived (which would be clec4f+/tim4-), so indeed we're probably majorly looking at EMP/Yolk sac derived cells when we believe the MS4a3 results (then expected to be tim4+clec4f+).

I'm not quite sure if I understand their comment on 3C/phenotype, I thought these scRNA-seq data were from indeed P0. On the zonation, looking at the few hepatocytes (N numbers would be helpful, comment #3) I am afraid the current dataset may not provide the power necessary for zonation analysis. Still it's an interesting idea, however not sure if required for this study's claims.

Last but not least, I fully agree to their comment on 4J, being in my view related to all my igf1 comments (#4,#6,#10,#11). I think this is exactly the point, and thus fuels my concern on the central role of the igf1 hypothesis (title and abstract) in the paper. I would be glad to learn about my co-reviewer's opinion on these points specifically.

Significance

My background is in liver development, liver metabolism, and single cell RNA sequencing, so my comments will primarily focus on these domains.

The authors provide an interesting study on the consequences on prenatal KC depletion on neonate mouse liver metabolism with the help of two conditional, RM-specific knockout models. Their findings show that prenatal KC depletion alters hepatic glycogen metabolism at birth. They suggest KC depletion is connected to accelerated hepatocyte maturation which is very interesting.

However, the study in its current shape remains rather descriptive. The interesting scRNA-seq data is not analyzed to the extent necessary to strengthen the claims on the putative central role of KC's post-depletion expression deficits on hepatocyte maturation and Igf1. Other cell types remain largely unstudied, although the data is present. Igf1, putatively secreted by KCs, is central to the claims of the study, yet mRNA or protein levels of Igf1 in KOSpi1 KCs are not presented, despite at least (sc)RNA-seq data at least for P0 are present. Other hormones central to glycogen metabolism, that could confound findings, such as glucagon, are not measured. KC-specific KO of Igf1 only moderately decreases liver Igf1 levels, and a transcriptional or metabolic characterization of hepatocytes in this KO (to test whether a phenotype similar to the KC-depleted mice) is missing. A KC-specific Igf1 rescue to confirm the importance of KC-derived Igf1 secretion for hepatocyte metabolism is not provided.

To conclude, the study provides valuable insights into murine hepatic metabolism at birth after prenatal KC depletion. Yet, to prove the claim that KCs directly control neonatal hepatocyte metabolism through Igf1, a number of additional experiments and controls would be needed.

Reviewer 2

Evidence, reproducibility and clarity

Using conditional knockout mouse models targeting macrophages, they demonstrate that yolk sac-derived KCs play a critical role in hepatocyte glycogen storage and function by regulating the tricarboxylic acid (TCA) cycle - a role that monocyte-derived KC-like cells cannot substitute. Newborn pups lacking yolk sac-derived KCs mobilize glycogen more rapidly, a process regulated by insulin-like growth factor 1 (Igf1) production.

Their findings reveal that macrophages are a major source of Igf1 at birth and that local Igf1 production by KCs is essential for balanced hepatocyte metabolism.

The paper is well structured with controlled-experiments that make the take-home message of the paper very clear and relevant.

Significance

The paper is novel and very relevant and support the importance of KC biology in liver homeostasis. However, some minor points should be addressed to strengthen the paper:

- To make the manuscript clearer. we advise authors to mainly focus their study using the spi1KO. Indeed, data from CSF1rKO macs look are 100% in agreement with spi1KO. Phenotype is different at birth (1G vs 1H).
- A better characterization of the Ms4a3Flp needs to be presented. Tomato goes from 1 to 6%. What does it mean? The 94% other does not come from HSC? Or bad efficiency? Considering the drastic depletion at E14.5, quite unexpected EMP-derived ones can repopulate, but might come from other sources? They comment this in the discussion but the main point is that data are inconclusive.
- Figure 1K, Tim4 upregulation is quite unexpected considering we are looking at HSC-derived KC-like cells? Any interpretation?
- Figure 3C, not sure of the monocle analysis on hepatocytes. The authors did not consider liver zonation in their analyses, any explanation for not considering it? What is the phenotype at P0?
- Figure 4J is supposed to consolidate the concept of the paper but we may expect a better outcome ($p > 0.01$).
- Discussion could be improved.

Suggestions:

What is the impact of having KC-like educating Hepatocytes perinatally in their mutants in the context of liver comorbidities or obesity?

Original submission

First decision letter

MS ID#: dev.204962

MS TITLE: Kupffer cells control neonatal hepatic metabolism via Igf1 signaling

AUTHORS: Nikola Makdissi, Daria J. Hirschmann, Aleksej Frolov, Inaam Sado, Fabian Nikolka, Jingyuan Cheng, Nelli Blank-Stein, Maria Francesca Viola, Mohamed Yaghmour, Philipp Arnold, Lorenzo Bonaguro, Matthias Becker, Christoph Thiele, Felix Meissner, Karsten Hiller, Marc D. Beyer and Elvira Mass

Dear Dr Mass,

Many thanks for transferring your paper to Development from Review Commons. I've looked at your manuscript, the reviews from Review Commons, and your revision plans. Reviewers seemed to be enthusiastic about the work and provided you some excellent feedback. Your revision plan looks very responsive to the comments and I would be delighted to consider a revised manuscript that includes the new data and revisions that you outline in your plan.

Please attend to all of the reviewers' comments and ensure that you clearly highlight all changes made in the revised manuscript. Please avoid using 'Tracked changes' in Word files as these are lost in PDF conversion. I should be grateful if you would also provide a point-by-point response detailing how you have dealt with the points raised by the reviewers in the 'Response to Reviewers' box. If you do not agree with any of their criticisms or suggestions please explain clearly why this is so.

First revision

Author response to reviewers' comments

Reviewer #1 (Evidence, reproducibility and clarity (Required)):

Summary:

Kupffer cells (KCs), the resident liver macrophages, inhabit the sinusoids of the liver where they serve as immune cells, defending against blood-borne pathogens. Their closest cellular neighbor are LSECs (Liver Sinusoidal Endothelial Cells), but they also interact with hepatocytes and stellate cells. KCs originate from hematopoietic cells. Initial hematopoiesis takes place in the blood islands of the yolk sac, from where precursors enter the blood stream and colonize the liver to form KCs. Yolk-sac derived KCs remain in the liver throughout life as an independent population. However, upon depletion, the KC pool can be replenished from circulating monocytes.

In their present study, the authors aim to investigate the role of resident KCs in the neonate mouse liver. To this end they deplete the prenatal resident macrophage population through a tissue-specific knock-out (targeting all pre-macrophages) of *Spi1* or *Csf1r*. In these models the prenatal KC population is near-depleted and by birth replenished largely by local precursors, rather than the circulating pool. They then compare the metabolic profile of wild-type and KO mice, and find metabolic deficits in the KOSpi1 livers (lowered glycogen). In scRNA-seq comparing WT and KOSpi1 mice, *Igf1* and *Igfr1* interactions of WT and KOSpi1 hepatocyte/KCs are reduced. They conclude that *Igf1* production is performed by yolk-sac derived, resident KCs, and that this *Igf1* is required for regulated hepatocyte glycogen metabolism in neonate mice.

Major comments:

1. Mouse models: small contribution of monocyte-derived KC-like cells to the total KC pool Only 5-7% of the KC population at P0 are derived by circulating monocytes, so the majority of impaired KC-like cells in *KOSpi1* is still yolk-sac derived. Therefore, the claim that yolk-sac origin is critical is a bit misleading, since they apparently only in minority contribute to the repopulation of the KC pool. If the authors wanted to keep their claim on the failure of the HSC-derived pool, they would need to separately characterize *Ms4a3*-positive KC-like cells in WT and *KOSpi1*, and compare them, demonstrating loss of marker expression in *Ms4a3*-positive *KOSpi1* KCs. Probably it is more useful to rephrase that the replenishment of the KCs is mainly driven by resident precursors, and therefore circulating replenishment is simply no option for restoration.

We have revised the manuscript to more clearly convey the main message: proper neonatal hepatocyte metabolism requires the co-development of hepatocytes with yolk sac-derived macrophages from the onset of liver organogenesis. Simply introducing KC-like cells at later developmental stages (i.e., after E14.5) is not sufficient to support normal metabolic function

To directly address the reviewer's point, we used the *Tnfrsf11a^{Cre}; Pu.1^{flox/flox}; Ms4a3^{FlpO}; Rosa26^{FSF-tdTomato}* model in combination with a high-dimensional flow cytometry panel. In this model we show that *tdTomato⁺* KC-like cells derived from *Ms4a3*-expressing progenitors behave exactly in the same way as *tdTomato⁻* KC-like cells in *KO^{Spi1}*, with the exception of *Clec2* expression (new Figure 1H). This new data demonstrates that the genetic deletion of KCs at E14.5 in *KO^{Spi1}* mice leads to a repopulation of the empty KC niche by different types of progenitors. Even though these *Ms4a3⁻* *tdTomato⁻* progenitors may originally stem from the yolk sac, at the time of depletion/repopulation these progenitors reside in the fetal liver. We have improved wording to convey this message better both in the results and discussion part (see lines 134-143, 394-398, 463-472).

Importantly, we would like to emphasize that during embryonic development, both EMP-derived pre-macrophages (pMacs) and GMP-derived monocytes are localized within the fetal liver. Therefore, no "circulating replenishment" of pMacs or monocytes is required at this stage. Only around E17.5-E18.5 do hematopoietic stem and progenitor cells begin to leave the liver and colonize the bone marrow niche.

2. Marker gene expression levels (Fig 1K)

It would be important to see expression levels of important KC markers *Clec2f*, *Clec4f*, *Crig* in the main figure.

We thank the reviewer for this suggestion. To address this and the previous point simultaneously, we assessed the expression of *Clec4f*, *Clec2*, and *Crig* (*Vsig4*) in the *Tnfrsf11a^{Cre}; Spi1^{fl/fl}* model crossed to *Ms4a3^{FlpO}; Rosa26^{FLF-tdTomato}* (*WT^{Spi1}; Ms4a3* and *KO^{Spi1}; Ms4a3*) to evaluate whether these established KC markers, known to play key roles in promoting KC identity in adult mice following depletion, are differentially expressed in KCs and KC-like cells (new Figure 1H, I). Interestingly, fetal and neonatal KCs do not necessarily follow the same differentiation trajectories as adult KCs, as not all KCs were *Vsig4⁺*, and *Tim4* expression was rather increased in KC-like cells compared to *bona fide* KCs. Nevertheless, this analysis was instrumental in determining that the newly incoming KC-like cells, independent of their origin, are phenotypically distinct from yolk sac-derived KCs.

3. Replicate numbers, n numbers, time points

Please state the n numbers (biol. replicates or how often the experiment was repeated), and the time point on top of every plot, or at least in the beginning of a new experiment e.g. when scRNAseq data is first introduced (annotate N, Ncells, timepoint). This will improve readability a lot.

This has been improved in the legends and figures, where applicable.

4. Igf1 expression reduced in *KOSpi1* or *Csf1r*? (qPCR or WB)

It would be very important to show whether *Igf1/Insr/Gcg* levels differ in E14.5/E12.5 and P0 KCs between WT/*KOSpi1* or *Csf1r* to keep the *Igf1* hypothesis in title and text. Maybe another cell population can be gated and compared as a control, to see whether we have cell-type specificity here. Additionally, the same expression profile would be expected in the already present RNA-Seq and scRNA-Seq data of KCs, and showing these as well would be needed to prove findings are consistent across methods.

To address this point, as well as point 9. raised by the same reviewer, which points in the same direction, we first extracted data from the developmental liver atlas by Qiu et al. (2024, Nature) to assess *Igf1* expression across liver-resident cell populations. This analysis clearly shows that KCs are the predominant producers of *Igf1* during development (new Figure 6A), establishing them as a key source of *Igf1* in the embryonic and early postnatal liver. Nevertheless, we have now expanded the Discussion to acknowledge that other sources of *Igf1* may also contribute to changes in hepatocyte metabolism (lines 449-452).

To further evaluate *Igf1* transcript dynamics, we re-analyzed *Igf1* expression across macrophage populations and their progenitors in various organs using the dataset from (Mass et al., 2016) (Figure R1). We observed that *Igf1* transcript levels increase upon differentiation into fetal liver (FL) macrophages (i.e. KCs) and continue to rise until postnatal day 21 (P21). For comparison, we also plotted *Igf1r* expression, which was found to be almost negligible relative to *Igf1* expression. This data together with the data from the single-cell atlas, reinforces that KCs are a key source of *Igf1* in the developing liver.

Figure R1: Heatmap representation of *Igf1* and *Igf1r* expression in sorted erythro-myeloid progenitors (EMPs), pre-macrophages (pMacs) and early macrophages (MF) at different developmental time points. FL: fetal liver; YS: yolk sac. Data from Mass et al., Science, 2016.

Consistently, a recent study showed that KCs are the main source of *Igf1* in the liver at E13.5 (Tang et al., 2025), which represents a time point when most KCs are depleted in both the *KO^{Spi1}* and *KO^{Csf1r}* models.

Building on this, we analyzed our bulk RNA-seq dataset of P0 *WT^{Spi1}* and *KO^{Spi1}* KCs, as well as sorted hepatocytes from the same pups. These analyses confirm that *WT^{Spi1}* KCs express slightly higher levels of *Igf1* than *KO^{Spi1}* KC-like cells (new Figure 5D), and that *Igf1* expression is substantially higher in KCs compared to hepatocytes. Despite the partial upregulation of *Igf1* observed in *KO^{Spi1}* KC-like cells at P0, the expression levels do not yet reach those of *WT^{Spi1}* KCs, suggesting that *Igf1* production during embryogenesis is insufficient to support proper hepatocyte maturation and function. We have added this point in the revised Discussion. Analysis of KC-derived factors, including *Igf1*, that are required for hepatocyte differentiation during embryogenesis is an ongoing project in the lab, therefore addressing this early KC-hepatocyte/hepatoblast crosstalk lies beyond the scope of the present study.

Along similar lines, and in response to the reviewer's specific suggestion to examine E12.5/E14.5 time points at the cellular level (e.g., by gating or sorting), we would like to

emphasize that KCs are largely absent in the knockout models at these stages. Consequently, testing Igf1 expression in KCs at these embryonic time points is not feasible, as the cell population of interest is essentially missing.

To validate the importance of KC-derived Igf1, we used the *Tnfrsf11a^{Cre}; Igf1^{fl/fl} (KO^{Igf1})* model, which depletes Igf1 starting at the pMac precursor stage. In these mice, both hepatic and circulating Igf1 levels are significantly reduced, confirming efficient gene deletion (this data was part of the original manuscript). Despite other cell types also producing Igf1, *KO^{Igf1}* animals show reduced hepatic glycogen content, supporting a role of macrophage-derived Igf1 in neonatal liver metabolism. We agree that additional Igf1 sources and paracrine factors may contribute to hepatocyte maturation, which could explain why the phenotype in *KO^{Igf1}* mice is less pronounced than in *KO^{Spi1}* and *KO^{Csf1r}* models. We have expanded on this point in the revised discussion.

To further address the reviewer's question, we quantified glucagon (Gcg) levels in the serum of *WT^{Spi1}* and *KO^{Spi1}* P0 pups and found no significant differences (new Figure 5F). Insulin serum levels, already included in the initial version of the manuscript (new Figure 5E), were likewise unchanged between genotypes. These data indicate that altered hepatocyte metabolism is not secondary to systemic changes in insulin or glucagon levels.

Regarding the reviewer's suggestion to validate Igf1 expression by qPCR or Western blot, we consider qPCR analyses redundant given that bulk and single-cell RNA-seq data already provide comprehensive transcript-level information. Western blot analysis also does not seem like a good option due to the little KC numbers that can be isolated from livers. Instead, we attempted to visualize Igf1 protein in KCs in P0 livers using the R&D

Systems AF791 antibody, which has been cited in previous studies. Since Tan et al. reported that signal amplification with a tyramide-based approach is required for this antibody, we used tyramide-AF647 (Thermo Fisher, B40958) according to the manufacturer's instructions. However, neither this protocol nor alternative conditions tested yielded a specific signal (**Figure R2**). The fluorescence signal that should represent Igf1 expression likely resulted from unspecific H₂O₂ conversion by heme present in red blood cells (see 2nd image without secondary antibody), which cannot be fully avoided because perfusion of P0 pups is technically unfeasible. Pretreatment of tissue sections with H₂O₂ completely abolished all Igf1 signal in both neonatal and adult liver (3rd and 4th picture in Figure R2), although adult hepatocytes should show high levels if Igf1 expression. We therefore refrained from testing additional antibodies, as similar staining artifacts were expected. While protein-level validation was not feasible at this stage, the transcriptional, genetic, and functional evidence presented supports a KC-specific, IGF1-dependent mechanism regulating hepatocyte function.

Figure R2: Immunofluorescence staining of P0 pups and adult livers. Representative confocal images showing nuclei stained with DAPI (gray), macrophages labeled with Iba1 (magenta), and immunoglobulin fragments (Igf1) detected in cyan. Scale bar: 50 µm.

Finally, we now include metabolic flux analyses demonstrating that Igf1 promotes pyruvate dehydrogenase (PDH) activity and thereby enhances TCA cycle flux specifically in *KO^{Igf1}* livers (new Figure 6J), further supporting a metabolic role of KC-derived Igf1 in the neonatal liver.

In summary, together with previous reports demonstrating Igf1 expression in macrophages (Yan et al., 2022; Rusin et al., 2024), the re-analyzed dataset shown in Figure new 6A and R1, and the data presented in the revised manuscript (new Figure 6), our findings reinforce that macrophage-derived Igf1 is an evolutionarily conserved mechanism that contributes substantially to tissue development and metabolic maturation in the neonatal liver.

5. Incomplete maturation of KOSpi1 KC-like cells may hinder proper monocyte recruitment
KOSpi1-KCs downregulate "genes such as Tnf, Il6, Il1a, Il1b, and Cxcl10 falling into the term 'inflammatory response' (Figure 1N)". However, we know that to open the niche and recruit monocytes, KCs have to release tumour necrosis factor (TNF) and IL-1 (Bonnardel, Immunity <https://doi.org/10.1016/j.immuni.2019.08.017>, 2019). May this indicate that insufficient restoration of KCs by circulating monocytes is potentially an artifact of the KOSpi1-KCs (and their reduced immune function)? This may be discussed in the text and statements on failed replenishment may be toned down.

We thank the reviewer for this insightful comment. The study by Bonnardel et al. (Immunity, 2019) indeed demonstrates that TNF and IL-1 are required to trigger niche opening and monocyte recruitment following KC depletion in the adult liver. However, in that context, these cytokines are primarily produced by hepatic stellate cells and endothelial cells, rather than by KCs themselves. Our bulk RNA-seq data, in contrast, focus exclusively on KCs.

Importantly, the KC niche at birth differs fundamentally from that in the adult liver. During the perinatal period, KCs are still establishing residence and migrate into hepatic sinusoids only about one week after birth (Araujo David et al., 2024). Consequently, the mechanisms that regulate KC differentiation and potential monocyte contribution during this developmental window are distinct from those described in the adult replenishment models.

In line with this, our flow cytometry data (Figure 1) indicate that fetal/neonatal KCs follow a distinct differentiation trajectory that does not fully recapitulate adult KC maturation (or activation) signatures. Therefore, the reduced expression of inflammatory genes in *KO^{Spi1}* KC-like cells likely reflects their immature developmental state rather than a defective inflammatory response.

6. Igf1 role - but what about glucagon levels?

The authors find lower liver glycogen levels at birth in *KOCsf1r* / *KOSpi1* mice, and I agree to their hypothesis this being in connection with metabolic dysregulation. However, glucagon levels in KOs could just be higher at birth and thus lead to more liver glycogen being mobilized. The authors even mention glucagon in the introduction but seem not to measure it. Assessment of glucagon levels or other important players of glycogen metabolism in P0 livers of WT vs *KOSpi1/Csf1r* would be an important control.

We have addressed this concern by measuring plasma glucagon levels in *KO^{Spi1}* mice at P0 to determine whether altered glucagon signaling contributes to the reduced liver glycogen content. ELISA analysis revealed no significant difference in glucagon concentrations between WT and KO animals (new Figure 5F), indicating that glucagon is unlikely to account for the observed metabolic phenotype.

7. ScRNAseq replicate numbers

It is not clear how many biological replicates contributed to the single cell data. I don't see a replicate integration in the methods, so I assume this is only one replicate which is probably not sufficient. Please provide more information and additional scRNA-seq replicates (usually 3).

The scRNA-seq dataset was generated from pooled liver cells of two pups per genotype. This information has now been added to the corresponding figure legend. We would like to

emphasize that the single-cell transcriptomic analysis was performed primarily as an exploratory approach to identify relevant cellular and molecular candidates, which we then validated at a genetic and functional level. Considering the high cost and resource intensity of scRNA-seq, and given that additional replicates would not significantly change or strengthen the main conclusions of the study, we believe that repeating the dataset is not necessary for the robustness of our findings.

8. Complex niche maturation deficit during embryonic development in KOs

Knockouts of *Spi1* and *Csf1r* are not temporally restricted, thus the metabolic deficits observed in the knockout livers could likewise result from any interference in liver niches (KC/LSEC/Hepatocyte/Cholangiocyte/HSC interaction) throughout prenatal development. E.g., the depletion of KCs could affect LSECs and only through LSECs hepatocytes or other related cell types. A thorough characterization of all the other cell types (e.g. LSECs, Stellate cells,...) including cell-cell interactions with hepatocytes in KO/WT livers should show no deficits compared to WT cells. Again, it would be important to show *Igf1* and *Insr* expression across all cell types. Importantly, cell-cell-interactions involving especially *Igf1* and *Igf1r* between all available cell types should be shown to rule out other cell types contributing. Until then, the assumption that impaired KC-like cells are the central contributor to hepatocyte dysregulation is speculative. Alternatively, a temporally- controlled depletion of resident KC's *Igf1* functionality at birth (once the niche has readily established) would be desirable, but probably more difficult.

We thank the reviewer for this interesting suggestion. As outlined above, the KC niche at birth is a completely different one than during adulthood. Cholangiocytes are still maturing and LSECs are not in such a frequent interaction with KCs as in the adult liver. E.g. we have previously described KC niches in the E14.5 liver where very few CD206+ KCs were associated with blood vessels (Kayvanjoo et al., 2024, eLife). Our data hints rather towards a KC-specific/*Igf1*-specific effect on hepatocyte maturation and function, as now assessed in more detail using the KO^{Igf1} model (new Figure 6).

We have also looked at the expression of *Igf1* in our scRNA-seq data set, however, the coverage of genes is quite sparse using the Seq-well method, which was the state-of-the-art for scRNA-seq when the experiments were performed and which offered loading of large cells, such as hepatocytes, onto nano-well arrays. Nevertheless, our data indicate that KCs are the main producers of *Igf1* transcripts (Figure R3). We have analyzed the bulk RNA-seq KC dataset at P0 from WT/KO^{Spi1} , as well as sorted hepatocytes from the same pups. These analyses show that WT^{Spi1} KC express higher slightly levels of *Igf1* than KO^{Spi1} KC-like cells. We have integrated this data now in the new Figure 5D.

Figure R3: Dotplot representation of expression from the scRNA-seq dataset from P0 livers.

Furthermore, to support the notion that *Igf1* is primarily produced by KCs during development, we analyzed all liver cells from the single-cell dataset published by Qiu et al. This analysis enabled us to visualize *Igf1* expression across hepatic cell types during development. As shown in the new Figure 6A, *Igf1* is predominantly expressed by KCs, reinforcing our experimental findings. To specifically address potential cell-cell interactions targeting the *Insr*, we analyzed Visfatin (*Nampt*) ligand activity using our scRNA-seq dataset (Figure R4) as the pair *Igf1*-*Insr* itself is not part of the CellChat database for ligand-receptor pairs and insulin is not expressed by hepatic cells. This analysis suggests that, in addition to KCs, megakaryocytes may also contribute to *Insr* activation in *WT^{Spi1}* livers. However, this effect is unlikely to involve increased *Igf1* production, as megakaryocytes do not express *Igf1* (see Figure R3). In contrast, *KO^{Spi1}* hepatocytes appear to receive no Visfatin-mediated input, further supporting our original observation that *Insr* signaling is impaired in these animals. Of note, our phosphoproteomics approach indicates that *Insr*, rather than *Igf1r*, is the main target of macrophage-derived *Igf1* (Table 4, lines 336-351).

WT*Spi1* VISFATIN signaling pathway network**KO*Spi1*** VISFATIN signaling pathway network
Figure R4: Circos plot of VISFATIN ligand activity using all cell types available in the scRNA-seq dataset.

Finally, at this point we do not believe that a “temporally-controlled depletion of resident KC’s Igf1 functionality at birth“, as suggested by the reviewer, is necessary to support our findings at this time. An in-depth analysis of KC-hepatoblast/hepatocyte crosstalk and how KCs may impact their differentiation, lies in our opinion beyond the scope of the present study and is currently under active investigation in the lab.

9. Validation of cells contributing to hepatocyte Igfr activation

Once the previous analysis has been performed and hints at KCs being the major source of Igf1, a staining for Igf1 (or probing the transcript) displaying co-localization with KCs (labeled with specific antibody) would be important to validate scRNA-seq observations.

See response to point 4.

10. Igf1 KO / Fig 4G-J

In figure 4G the reduction if liver Igf1 is used to validate the KC-Igf1 KO, but actually it is not so dramatic, hinting at other Igf1 sources. The statement “We first confirmed that macrophages were the primary producers of Igf1 at P0.” needs to be toned down urgently, looking at 4G I can immediately count at least 4 dots with exactly the same Igf1 level as in the WT. Again the glycogen (J) reduction looks very moderate. The strong statement on Igf1 role again may need to be toned down. Alternatively, a rescue experiment restoring hepatocyte transcriptional/metabolic deficits and glycogen storage upon Igf1 delivery may be provided.

Although Igf1 is produced by many different cell types, our findings show that - on average - at least one third of hepatic and up to half of circulating Igf1 protein levels in newborn mice are macrophage-derived, indicating that this cell type is a significant source of Igf1 at this developmental stage. Nevertheless, we agree that the wording may have overstated this conclusion, and we have revised the text to reflect the findings more precisely (lines 359-361).

We appreciate the suggestion of a rescue experiment. However, since KC depletion occurs between embryonic days E12.5 and E14.5, such an experiment would require repeated in utero injections of recombinant Igf1 protein or the generation of a new mouse line allowing inducible Igf1 overexpression (e.g., via tamoxifen administration). In our view, both approaches are

technically demanding and beyond the scope of the current manuscript - particularly given that we already employ a conditional *Igf1* knockout model that provides mechanistic insight. To strengthen the already presented significant reduction of glycogen using a colorimetric assay, we additionally quantified the glycogen content of hepatocytes using transmission electron microscopy (new Figure 6F-H).

Furthermore, we performed metabolic flux analyses in *WT^{Igf1}* and *KO^{Igf1}* livers with or without the addition of recombinant *Igf1* to the culture medium to assess the direct effects of *Igf1* on neonatal hepatocyte metabolism (new Figure 6J). These experiments reinforce the requirement for KC-derived *Igf1*, as only *KO^{Igf1}* livers responded to exogenous *Igf1* by enhancing pathways associated with pyruvate dehydrogenase (PDH) activity, whereas WT livers, already exposed to endogenous *Igf1*, did not show further stimulation upon *Igf1* addition.

11. Title/Scope

The metabolic and transcriptomic characterization of neonate *KO^{Spi1}* liver is interesting and thorough. Based on the findings, the association with the *Igf1* phenotype without rescue experiments is less strong. Maybe the title can be toned down accordingly, e.g. stating "Characterization of hepatic metabolism and *Igf1* signaling in two prenatal KC depletion mouse models"

As discussed above, a rescue experiment is beyond the scope of the current manuscript. However, the use of the conditional *Igf1* knockout model allowed us to provide in-depth mechanistic insight into the role of macrophage-derived *Igf1*. Given the new results of the *KO^{Igf1}* model, we would prefer to retain the current title of the manuscript.

Minor comments:

1. Quantification plots (e.g. 1C,G,H,J,L...) would benefit from adding the time point in the title. Please add time points for all figures to help readability.

We added the suggested information to the figures. The timepoint has been indicated if more than one developmental timepoint was analyzed in the respective figure.

2. Some of the references are not formatted correctly

We corrected this, if here the DOI was meant.

3. The reduction in PAS signal in Fig 2 doesn't look very convincing to me, also it is not significant (p0.07). Please change the text accordingly.

Although PAS quantification did not reach statistical significance, it indicated a clear trend toward reduced glycogen levels. To substantiate this observation, we performed more precise biochemical assays and imaging analyses, included already in the original submission, which confirmed and strengthened our initial findings by demonstrating significantly reduced hepatic glycogen storage in *KO^{Spi1}* mice. We therefore consider the current wording - "Quantification of the staining indicated that *Spi1*KO livers stored less glycogen than their littermate controls" - to be appropriate and accurate, particularly in light of the data presented in Figure 3D, E.

4. There is two Figure 4

We corrected this.

5. Fig.4 G/H says *Igf1* KO, but in the legend *WT^{Spi1}* and *KO^{Spi1}* at P0. For interpretation this is important, please also check the other legends.

We corrected this.

****Referees cross-commenting****

I appreciate my co-reviewer's comments.

Their suggestion to focus on spi1KO makes sense. Still I would suggest keeping the information on csf1r in the supplements. They are indeed different at birth, this information may be valuable for some readers, additionally more models may make the paper more likely to be cited.

We thank the reviewer for this comment, and moved the data related to the Csf1r model to the supplements.

I completely agree to their comment on the Ms4a3, I see this related to my comments #1 and #5.

Their comment on Tim4 makes sense, I see this in line with my request for more KC markers #2, especially Clec4f. I would add that these cells are still only 5% HSC derived (which would be clec4f+/tim4-), so indeed we're probably majorly looking at EMP/Yolk sac derived cells when we believe the MS4a3 results (then expected to be tim4+clec4f+).

I'm not quite sure if I understand their comment on 3C/phenotype, I thought these scRNA-seq data were from indeed P0. On the zonation, looking at the few hepatocytes (N numbers would be helpful, comment #3) I am afraid the current dataset may not provide the power necessary for zonation analysis. Still it's an interesting idea, however not sure if required for this study's claims.

Last but not least, I fully agree to their comment on 4J, being in my view related to all my igf1 comments (#4,#6,#10,#11). I think this is exactly the point, and thus fuels my concern on the central role of the igf1 hypothesis (title and abstract) in the paper. I would be glad to learn about my co-reviewer's opinion on these points specifically.

Reviewer #1 (Significance (Required)):

My background is in liver development, liver metabolism, and single cell RNA sequencing, so my comments will primarily focus on these domains.

The authors provide an interesting study on the consequences on prenatal KC depletion on neonate mouse liver metabolism with the help of two conditional, RM-specific knockout models. Their findings show that prenatal KC depletion alters hepatic glycogen metabolism at birth. They suggest KC depletion is connected to accelerated hepatocyte maturation which is very interesting.

However, the study in its current shape remains rather descriptive. The interesting scRNA-seq data is not analyzed to the extent necessary to strengthen the claims on the putative central role of KC's post-depletion expression deficits on hepatocyte maturation and Igf1.

Other cell types remain largely unstudied, although the data is present. Igf1, putatively secreted by KCs, is central to the claims of the study, yet mRNA or protein levels of Igf1 in KOSpi1 KCs are not presented, despite at least (sc)RNA-seq data at least for P0 are present. Other hormones central to glycogen metabolism, that could confound findings, such as glucagon, are not measured. KC-specific KO of Igf1 only moderately decreases liver Igf1 levels, and a transcriptional or metabolic characterization of hepatocytes in this KO (to test whether a phenotype similar to the KC-depleted mice) is missing. A KC-specific Igf1 rescue to confirm the importance of KC-derived Igf1 secretion for hepatocyte metabolism is not provided.

To conclude, the study provides valuable insights into murine hepatic metabolism at birth after prenatal KC depletion. Yet, to prove the claim that KCs directly control neonatal hepatocyte metabolism through Igf1, a number of additional experiments and controls would be needed.

We thank the reviewer for the overall positive feedback. Respectfully, we disagree with the characterization of our study as merely descriptive. In addition to phenotypic observations, we

provide metabolic flux analysis and phosphoproteomic data, which offer direct insights into active signaling pathways and TCA cycle dynamics. These approaches go beyond descriptive work and contribute mechanistic understanding. Nevertheless, we appreciate the reviewer's suggestions and have carried out additional experiments, including glucagon measurements and a more detailed investigation into the role of Igf1 in KC-hepatocyte crosstalk, to further strengthen the mechanistic basis of our findings. We trust that our responses to this reviewer and reviewer #2 clarify the points raised and resolve the remaining concerns.

Reviewer #2 (Evidence, reproducibility and clarity (Required)):

Using conditional knockout mouse models targeting macrophages, they demonstrate that yolk sac-derived KCs play a critical role in hepatocyte glycogen storage and function by regulating the tricarboxylic acid (TCA) cycle - a role that monocyte-derived KC-like cells cannot substitute.

Newborn pups lacking yolk sac-derived KCs mobilize glycogen more rapidly, a process regulated by insulin-like growth factor 1 (Igf1) production.

Their findings reveal that macrophages are a major source of Igf1 at birth and that local Igf1 production by KCs is essential for balanced hepatocyte metabolism.

The paper is well structured with controlled-experiments that make the take-home message of the paper very clear and relevant.

We thank the reviewer for the positive assessment of our work and have included additional data to strengthen our conclusions.

Reviewer #2 (Significance (Required)):

The paper is novel and very relevant and support the importance of KC biology in liver homeostasis. However, some minor points should be addressed to strengthen the paper:

- To make the manuscript clearer. we advise authors to mainly focus their study using the spi1KO. Indeed, data from CSF1rKO macs look are 100% in agreement with spi1KO. Phenotype is different at birth (1G vs 1H).

In response to this comment and that of Reviewer #1, we have moved all data related to the Csf1r KO model to the Supplementary Figures. While this model exhibits distinct dynamics compared to the Spi1 KO model, the main phenotypes are consistent across both, thereby reinforcing and supporting our overall conclusions.

- A better characterization of the Ms4a3Flp needs to be presented. Tomato goes from 1 to 6%. What does it mean? The 94% other does not come from HSC? Or bad efficiency? Considering the drastic depletion at E14.5, quite unexpected EMP-derived ones can repopulate, but might come from other sources? They comment this in the discussion but the main point is that data are inconclusive.

We now present the phenotypic characterization of tdTomato⁺ versus tdTomato⁻ KCs/KC-like cells in the new Figure 1H, I demonstrating that these populations are phenotypically distinct. The model itself is highly efficient, with labeling of approximately 80% of Ly6C⁺ monocytes at P0 (new Figure 1G).

When assessing the recombination efficiency of the Ms4a3-Flp system at E14.5, which is the peak timepoint of KC depletion, we observe labeling in only ~20% of GMPs, ~30% of Ly6C⁺ Cx3cr1⁻ monocytes, and less than 5% of Ly6C⁺ Cx3cr1⁺ monocytes (Figure R5A). These data suggest that rather Ly6C⁺ Cx3cr1⁺ monocytes are repopulating the empty KC niche as the % of tdTomato labelling is similar to what we observe at P0. However, the origin of tdTomato⁻ Ly6C⁺ Cx3cr1⁺ monocytes remains unclear. Recent studies suggest long-term HSC-independent hematopoiesis in the fetal liver (Kobayashi et al., 2023; Yokomizo et al., 2022), which may partially account for the tdTomato⁻ populations of these transient hematopoietic waves in our m

Interestingly, in other organs such as the brown adipose tissue (BAT), replenishment by $tdTomato^+$ cells in the $KO^{Spi1}; Ms4a3$ models appears more efficient (Figure R5B), suggesting that GMP-derived cells may be intrinsically limited in their ability to repopulate the KC niche.

Figure R5: (A) Fate mapping of $tdTomato^+$ cells in $Ms4a3^{Flp}; Rosa26^{FSF-tdTomato}$ E14.5 fetal livers (FL). GMP: granulocyte-monocyte progenitor. Monocytes were split into $Ly6C^+ Cx3cr1^-$ and $Ly6C^+ Cx3cr1^+$ populations (see also Kayvanjoo et al. 2024). Macrophages were defined as $F4/80^{high} CD11b^{int}$. **(B)** Quantification of brown adipose tissue (BAT) macrophages at P0 in WT^{Spi1} and KO^{Spi1} combined with the $Ms4a3^{Flp}; Rosa26^{FSF-tdTomato}$ mouse model. See Figure 1F for breeding scheme. Circles indicate biological replicates. Unpaired Student's t-test.

As discussed in the manuscript, the precise origin of KC-like cells remains elusive. Nonetheless, this limitation does not undermine our central conclusion: that proper hepatocyte function during development critically depends on co-development with early KCs, and that disruption of this interaction leads to impaired hepatic metabolic function.

- Figure 1K, Tim4 upregulation is quite unexpected considering we are looking at HSC-derived KC-like cells? Any interpretation?

We have re-analyzed our data comparing $tdTomato^+$ versus $tdTomato^-$ KCs/KC-like cells. As discussed in the manuscript and outlined above, it is unlikely that definitive HSC- or GMP-derived cells differentiate efficiently into KC-like cells in this context. Therefore, direct comparisons to studies involving adult or postnatal KC depletion are limited. To better define the characteristics of these cells, we analyzed additional KC and KC-like markers in $WT^{Spi1}; Ms4a3$ and $KO^{Spi1}; Ms4a3$ animals (see response to Reviewer #1, points 1+2), allowing more precise conclusions regarding their phenotype (new Figure 1G, H).

Interestingly, even in the fate-mapper model combined with the $Spi1$ knockout, the repopulating, freshly incoming macrophages displayed higher levels of Tim4 expression. Although unexpected, this finding reinforces that the fetal and neonatal liver microenvironment differs profoundly from that of the adult liver. In this developmental setting, Tim4 does not serve as a marker of long-term tissue residency or yolk-sac origin but rather reflects an early macrophage differentiation state shaped by the unique perinatal niche.

Consistent with this interpretation, KCs acquire adult-like features only about one week after birth, once they have migrated from the parenchyma into the hepatic sinusoids (Araujo David et al., 2024). We therefore consider it likely that Tim4 expression is maintained at this stage mainly by yolk-sac- or fetal-liver-derived macrophages, which, independent of their precise origin, transiently express Tim4 during this developmental

transition. In line with our previous observations (Mass et al., 2016), Tim4 should thus be regarded as a general marker of macrophage lineage commitment, rather than a definitive indicator of mature, tissue-resident KC identity during early liver development.

- Figure 3C, not sure of the monocle analysis on hepatocytes. The authors did not consider liver zonation in their analyses, any explanation for not considering it? What is the phenotype at P0?

There is no hepatocyte zonation at p0 yet, therefore we did not perform this analysis. We have added this point to the discussion for clarification.

- Figure 4J is supposed to consolidate the concept of the paper but we may expect a better outcome ($p > 0.01$).

We agree with the reviewer and have performed the following experiments to strengthen our manuscript:

1. Transmission electron microscopy on the KO^{Igf1} model, which confirmed reduced glycogen levels, similar to our observations in the $KOSpi1$ model (new Figure 6F, G, H).
2. We performed metabolic flux analyses demonstrating that Igf1 promotes pyruvate dehydrogenase (PDH) activity and thereby enhances TCA cycle flux specifically in KO^{Igf1} livers (new Figure 6J), further supporting a metabolic role of KC-derived Igf1 in the neonatal liver.

- Discussion could be improved.

We improved the discussion to incorporate all findings and limitations.

Suggestions:

What is the impact of having KC-like educating Hepatocytes perinatally in their mutants in the context of liver comorbidities or obesity?

This is an important and insightful point raised by the reviewer. Unfortunately, both KO^{Spi1} and KO^{Csf1r} mice die within days to weeks after birth, which precludes the possibility of performing long-term studies in these models. However, we are actively exploring alternative and more precise strategies for KC depletion in the perinatal period.

For instance, we have attempted to use the *Clec4f-Cre* line crossed to $Spi1^{flox}$ animals. However, this approach did not result in effective depletion of KCs before birth, as *Clec4f*-driven Cre expression initiates only around E17.5. We are currently analyzing this model with a focus on early postnatal hepatocyte metabolism, as we observe KC depletion at around 2 weeks of age, followed by partial replenishment by KC-like cells.

While these ongoing efforts are promising, we believe that a detailed mechanistic investigation of these models is beyond the scope of the current manuscript. Nonetheless, they represent an important direction for future studies.

References:

- Araujo David, B., J. Atif, F. Vargas E Silva Castanheira, T. Yasmin, A. Guillot, Y. Ait Ahmed, M. Peiseler, J.W. Hommes, L. Salm, M.A. Brundler, B.G.J. Surewaard, W. Elhenawy, S. MacParland, F. Ginhoux, K. McCoy, and P. Kubes. 2024. Kupffer cell reverse migration into the liver sinusoids mitigates neonatal sepsis and meningitis. *Sci Immunol*. 9:eadq9704. doi:10.1126/SCIIMMUNOL.ADQ9704.
- Kayvanjoo, A.H., I. Splichalova, D.A. Bejarano, H. Huang, K. Mael, N. Makdissi, D. Heider, H.M. Tew, N.R. Balzer, E. Greto, C. Osei-Sarpong, K. Baßler, J.L. Schultze, S. Uderhardt, E. Kiermaier, M. Beyer, A. Schlitzer, and E. Mass. 2024. Fetal liver macrophages contribute to the hematopoietic stem cell niche by controlling granulopoiesis. *Elife*. 13. doi:10.7554/ELIFE.86493.
- Kobayashi, M., H. Wei, T. Yamanashi, N. Azevedo Portilho, S. Cornelius, N. Valiente, C. Nishida, H. Cheng, A. Latorre, W.J. Zheng, J. Kang, J. Seita, D.J. Shih, J.Q. Wu, and M. Yoshimoto. 2023. HSC-independent definitive hematopoiesis persists into adult life. *Cell*

- Rep.* 42:112239. doi:10.1016/J.CELREP.2023.112239.
- Mass, E., I. Ballesteros, M. Farlik, F. Halbritter, P. Gunther, L. Crozet, C.E. Jacome-Galarza, K. Handler, J. Klughammer, Y. Kobayashi, E. Gomez-Perdiguero, J.L. Schultze, M. Beyer, C. Bock, and F. Geissmann. 2016. Specification of tissue-resident macrophages during organogenesis. *Science (1979)*. 353:aaf4238-aaf4238. doi:10.1126/science.aaf4238.
- Rusin, D., L. Vahl Becirovic, G. Lyszczyk, M. Krueger, A. Benmamar-Badel, C. Vad Mathiesen, E. Sigurðardóttir Schiöth, K. Lykke Lambertsen, and A. Włodarczyk. 2024. Microglia-Derived Insulin-like Growth Factor 1 Is Critical for Neurodevelopment. *Cells*. 13. doi:10.3390/CELLS13020184.
- Tang, X.T., L.V. Chen, and B.O. Zhou. 2025. Resolving the spatial organization of fetal liver hematopoiesis by SeekSpace. *Cell Regeneration 2025 14:1*. 14:1-15. doi:10.1186/S13619-025-00234-0.
- Yan, X., E. Managlia, Y.Y. Zhao, X. Di Tan, and I.G. De Plaen. 2022. Macrophage-derived IGF-1 protects the neonatal intestine against necrotizing enterocolitis by promoting microvascular development. *Communications Biology 2022 5:1*. 5:1-13. doi:10.1038/s42003-022-03252-9.
- Yokomizo, T., T. Ideue, S. Morino-Koga, C.Y. Tham, T. Sato, N. Takeda, Y. Kubota, M. Kurokawa, N. Komatsu, M. Ogawa, K. Araki, M. Osato, and T. Suda. 2022. Independent origins of fetal liver haematopoietic stem and progenitor cells. *Nature 2022 609:7928*. 609:779-784. doi:10.1038/s41586-022-05203-0.

Second decision letter

MS ID#: dev.204962R1

MS TITLE: Kupffer cells control neonatal hepatic metabolism via Igf1 signaling

AUTHORS: Nikola Makdissi, Daria J. Hirschmann, Aleksej Frolov, Inaam Sado, Bastian Bennühr, Fabian Nikolka, Jingyuan Cheng, Nelli Blank-Stein, Maria Francesca Viola, Mohamed Yaghmour, Philipp Arnold, Lorenzo Bonaguro, Matthias Becker, Christoph Thiele, Felix Meissner, Karsten Hiller, Marc D. Beyer and Elvira Mass

Dear Dr Mass,

I am happy to tell you that your manuscript has been accepted for publication in *Development*, pending our standard publication integrity checks.

Reviewer 1: I thank the authors for their efforts in addressing my points in the realm of the feasible within the scope of their current project.

Specifically,

- * the FACS-characterization of KC/KC-like cells is interesting, clarifying that rather than yolk-sac origin, complex niche alterations may explain the observed phenotype, and the new discussion reflects this nicely.
- * the addition of glucagon measurements is helpful to rule out systemic confounders.
- * additional validation experiments on reduced glycogen content and response to exogenous Igf1 in Figure 6 strengthen the Igf1 focus in the study
- * the new Figure 5D is important to show that effect sizes are small at P0, prompting efforts to study interactions during embryogenesis in closer detail
- * in line with this the adjusted discussion acknowledges the potential contribution of other liver cells and paracrine factors

Overall, I am pleased with the state of the manuscript and have no further comments.

Reviewer 2

Advance summary and potential significance to field

The role of liver-derived paracrine factors in neonatal hepatocyte metabolism is poorly understood and is the focus of this study. In this revised report, the authors include new data which further support and strengthen their hypothesis that yolk-sac-derived Kupffer cells (KC) specifically and, KC-derived factor, Igf-1, in particular, are major contributors to the development and metabolic function of neonatal (P0) hepatocytes. Substantial experimental evidence provided (transcriptional landscape, metabolic tracing, mitochondrial respiration, lipid analyses) support requirement of yolk-sac-derived KCs for proper neonatal hepatocyte metabolic function at birth which cannot be fully compensated by KC-like cells such as GMP-derived KC (Figures 1-4 plus supplemental data).

The identification of Igf-1 signaling pathway as a key mediator in yolk-sac-derived KC-hepatocyte cross-talk necessary for appropriate hepatocyte metabolism at birth, is intriguing (Figures 5-6). The demonstration that in KC-specific KOIgf1, the observed disruption in hepatic metabolic flux but could be reorientated towards oxidative pathways by reintroduction of Igf1, convincingly support role of Igf-1 signaling in neonatal hepatic metabolic homeostasis. However, the milder phenotype observed in KC-specific KOIgf1 compared to KC-derived KOSpi1, led the authors to acknowledge that KC-derived Igf1 most likely fine-tunes the balance between hepatic glycolytic and oxidative metabolism as opposed to driving total carbon flux, and that additional paracrine factors are likely involved as part of orchestrated efforts.

Overall, the authors have convincingly and logically demonstrated a new critical role of yolk-sac-derived KC for developmentally regulating neonatal hepatic metabolic homeostasis via yolk-sac KC-hepatocyte cross-talk involving Igf1 signaling. Previous concerns were addressed, and data interpretation, including limitations, very well discussed.

Comments for the author

Although not necessary for present report, generation and evaluating hepatocyte-specific knock-out of Igf1r would lend support for importance of Igf1 signaling in KC-hepatocyte cross-talk at P0. Of note, Igf-1 has low affinity for Insr.

Reviewer 3: The authors have thoroughly addressed all of my concerns. I congratulate them on their careful revision